# A COMPUTATIONAL FRAMEWORK FOR SOLVING WASSERSTEIN LAGRANGIAN FLOWS

## ABSTRACT

The dynamical formulation of the optimal transport can be extended through various choices of the underlying geometry (*kinetic energy*), and the regularization of density paths (*potential energy*). These combinations yield different variational problems (*Lagrangians*), encompassing many variations of the optimal transport problem such as the Schrödinger Bridge, unbalanced optimal transport, and optimal transport with physical constraints, among others. In general, the optimal density path is unknown, and solving these variational problems can be computationally challenging. Leveraging the dual formulation of the Lagrangians, we propose a novel deep learning based framework approaching all of these problems from a unified perspective. Our method does not require simulating or backpropagating through the trajectories of the learned dynamics, and does not need access to optimal couplings. We showcase the versatility of the proposed framework by outperforming previous approaches for the single-cell trajectory inference, where incorporating prior knowledge into the dynamics is crucial for correct predictions.

## 1 INTRODUCTION

The problem of *trajectory inference*, or recovering the population dynamics of a system from samples of its temporal marginal distributions, is a problem arising throughout the natural sciences (Hashimoto et al., 2016; Lavenant et al., 2021). A particularly important application is analysis of single-cell RNA-sequencing data (Schiebinger et al., 2019; Schiebinger, 2021; Saelens et al., 2019), which provides a heterogeneous snapshot of a cell population at a high resolution, allowing high-throughput observation over tens of thousands of genes (Macosko et al., 2015). However, since the measurement process ultimately leads to cell death, we can only observe temporal changes of the *marginal* or *population* distributions of cells as they undergo treatment, differentiation, or developmental processes of interest. To understand these processes and make future predictions, we are interested in both (i) interpolating the evolution of marginal cell distributions between observed timepoints and (ii) modeling the full trajectories at the individual cell level.

However, when inferring trajectories over cell distributions, there exist multiple cell dynamics that yield the same population marginals. This presents an ill-posed problem, which highlights the need for trajectory inference methods to be able to flexibly incorporate different types of prior information on the cell dynamics. Commonly, such prior information is specified via posing a variational problem on the space of marginal distributions, where previous work on measure-valued splines (Chen et al., 2018; Benamou et al., 2019; Chewi et al., 2021; Clancy & Suarez, 2022; Chen et al., 2023) are examples which seek minimize the acceleration of particles.

We propose a general framework for using deep neural networks to infer dynamics and solve marginal interpolation problems, using Lagrangian action functionals on manifolds of probability densities that can flexibly incorporate various types of prior information. We consider Lagrangians of the form $\mathcal{L}[\rho_t, \dot{\rho}_t, t] = \mathcal{K}[\rho_t, \dot{\rho}_t, t] - \mathcal{U}[\rho_t, t]$, referring to the first term as a *kinetic energy* and the second as a *potential energy*. Our methods can be used to solve a diverse family of problems defined by the choice of these energies and constraints on the evolution of $\rho_t$. More explicitly, we specify

- A *kinetic energy* which, in the primary examples considered in this paper, corresponds to a *geometry* on the space of probability measures. We primarily consider the Riemannian structures corresponding to the Wasserstein-2 and Wasserstein Fisher-Rao metrics.

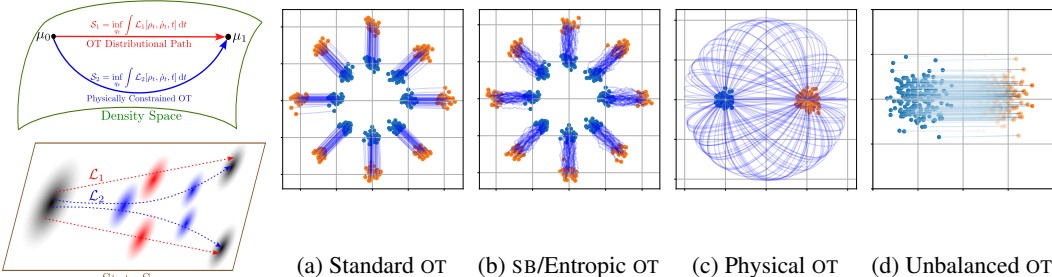

(a) Standard OT  (b) SB/Entropic OT  (c) Physical OT  (d) Unbalanced OT

Figure 1: Our *Wasserstein Lagrangian Flows* are action-minimizing curves for various choices of Lagrangian $\mathcal{L}_i[\rho_t, \dot{\rho}_t, t]$ on the space of densities, which each translate to particular state-space dynamics. Toy examples of dynamics resulting from various potential or kinetic energy terms are given in (a)-(d). We may also constrain Wasserstein Lagrangian flows match intermediate data marginals $\rho_{t_i} = \mu_{t_i}$ and combine energy terms to define a suitable notion of interpolation between given $\mu_{t_i}$.

- A *potential energy*, which is a functional of the density, for example the expectation of a physical potential encoding prior knowledge or even a nonlinear functional.
- A collection of *marginal* constraints which are inspired by the availability of data in the problem of interest. For optimal transport (OT), Schrödinger Bridge (SB), or generative modeling tasks, we are often interested in interpolating between two endpoint marginals given by a data distribution and/or a tractable prior distribution. For applications in trajectory inference, we may incorporate multiple constraints to match the observed temporal marginals, given via data samples. Notably, in the limit of data sampled infinitely densely in time, we recover the Action Matching (AM) framework of Neklyudov et al. (2023).

Within our *Wasserstein Lagrangian Flows* framework, we propose tractable dual objectives to solve (i) *standard Wasserstein-2* OT (Ex. 4.1, Benamou & Brenier (2000); Villani (2009)), (ii) *entropy regularized* OT or Schrödinger Bridge (Ex. 4.4, Léonard (2013); Chen et al. (2021b), (iii) *physically constrained* OT (Ex. 4.3, Tong et al. (2020); Koshizuka & Sato (2022)), and (iv) *unbalanced* OT (Ex. 4.2, Chizat et al. (2018a)) (Sec. 4). Our framework also allows for combining energy terms to incorporate features of the above problems as inductive biases for trajectory inference. In Sec. 5, we showcase the ability of our methods to accurately solve Wasserstein Lagrangian flow optimizations, and highlight how testing different Lagrangians can improve results in single-cell RNA-sequencing applications. We discuss benefits of our approach compared to related work in Sec. 6.

## 2 BACKGROUND

### 2.1 WASSERSTEIN-2 GEOMETRY

For two given densities with finite second moments $\mu_0, \mu_1 \in \mathcal{P}_2(\mathcal{X})$, the Wasserstein-2 OT problem is defined, in the Kantorovich formulation, as a cost-minimization problem over joint distributions or 'couplings' $\pi \in \Pi(\mu_0, \mu_1) = \{\pi(x_0, x_1) \mid \int \pi(x_0, x_1) dx_1 = \mu_0, \int \pi(x_0, x_1) dx_0 = \mu_1\}$, i.e.

$$W_2(\mu_0, \mu_1)^2 := \inf_{\pi \in \Pi(\mu_0, \mu_1)} \int \|x_0 - x_1\|^2 \pi(x_0, x_1) dx_0 dx_1 . \tag{1}$$

The dynamical formulation of Benamou & Brenier (2000) gives an alternative perspective on the $W_2$ OT problem as an optimization over a vector field $v_t$ that transports samples according to an ODE $\dot{x}_t = v_t$. The evolution of the samples' density $\rho_t$, under transport by $v_t$, is governed by the *continuity equation* $\dot{\rho}_t = -\nabla \cdot (\rho_t v_t)$ (Figalli & Glaudo (2021) Lemma 4.1.1), and we have

$$W_2(\mu_0, \mu_1)^2 = \inf_{\rho_t} \inf_{v_t} \int_0^1 \int \frac{1}{2} \|v_t\|^2 \rho_t \, dx_t dt \quad \text{s.t. } \dot{\rho}_t = -\nabla \cdot (\rho_t v_t), \ \rho_0 = \mu_0, \ \rho_1 = \mu_1, \tag{2}$$

where $\nabla \cdot ()$ is the divergence operator. The $W_2$ transport cost can be viewed as providing a Riemannian manifold structure on $\mathcal{P}_2(\mathcal{X})$ (Otto (2001); Ambrosio et al. (2008), see also Figalli & Glaudo (2021) Ch. 4). Introducing Lagrange multipliers $\phi_t$ to enforce the constraints in Eq. (2), we obtain the condition $v_t = \nabla \phi_t$ (see App. B.1), which is suggestive of the result from Ambrosio et al. (2008) characterizing the tangent space $T_\rho^{W_2} \mathcal{P}_2 = \{\dot{\rho} \mid \int \dot{\rho} \, dx_t = 0\}$ via the continuity equation,

$$T_\rho^{W_2} \mathcal{P}_2(\mathcal{X}) = \{\dot{\rho} \mid \dot{\rho} = -\nabla \cdot (\rho \nabla \phi)\}. \tag{3}$$

We also write the cotangent space as $T_\rho^{*W_2} \mathcal{P}_2(\mathcal{X}) = \{[\phi] \mid \phi \in \mathcal{C}^\infty(\mathcal{X})\}$, where $\mathcal{C}^\infty(\mathcal{X})$ denotes smooth functions and $[\phi]$ is an equivalence class up to addition by a constant. For two curves $\mu_t, \rho_t : [-\epsilon, \epsilon] \mapsto \mathcal{P}_2(\mathcal{X})$ passing through $\rho := \rho_0 = \mu_0$, the Otto metric is defined

$$\langle \dot{\mu}_t, \dot{\rho}_t \rangle_{T_\rho}^{W_2} = \langle \phi_{\dot{\mu}_t}, \phi_{\dot{\rho}_t} \rangle_{T_\rho^*}^{W_2} = \int \langle \nabla \phi_{\dot{\mu}_t}, \nabla \phi_{\dot{\rho}_t} \rangle \rho \, dx. \tag{4}$$

## 2.2 WASSERSTEIN FISHER-RAO GEOMETRY

Building from the dynamical formulation in Eq. (2), Chizat et al. (2018a;b); Kondratyev et al. (2016); Liero et al. (2016; 2018) consider additional terms allowing for birth and death of particles, or teleportation of probability mass. In particular, consider extending the continuity equation to include a 'growth term' $g_t : \mathcal{X} \to \mathbb{R}$ whose norm is regularized in the cost,

$$WFR_\lambda(\mu_0, \mu_1)^2 = \inf_{\rho_t} \inf_{v_t, g_t} \int_0^1 \int \left( \frac{1}{2} \|v_t\|^2 + \frac{\lambda}{2} g_t^2 \right) \rho_t \, dx_t dt, \tag{5}$$

subject to $\dot{\rho}_t = -\nabla \cdot (\rho_t v_t) + \lambda \rho_t g_t, \rho_0 = \mu_0, \rho_1 = \mu_1$. We call this the Wasserstein Fisher-Rao (WFR) distance, since considering *only* the growth terms recovers the non-parametric Fisher-Rao metric (Chizat et al., 2018a; Bauer et al., 2016). We also refer to Eq. (5) as the *unbalanced* OT problem on the space of unnormalized densities $\mathcal{M}(\mathcal{X})$, since the growth terms need not preserve normalization $\int \dot{\rho}_t dx_t = \int \lambda g_t \rho_t dx_t \neq 0$ without further modifications (see e.g. Lu et al. (2019)).

Kondratyev et al. (2016) define a Riemannian structure on $\mathcal{M}(\mathcal{X})$ via the WFR distance. Introducing Lagrange multipliers $\phi_t$ and eliminating $v_t, g_t$ in Eq. (5) yields the optimality conditions $v_t = \nabla \phi_t$ and $g_t = \phi_t$. In analogy with Sec. 2.1, this suggests characterizing the tangent space via the tuple $(\phi_t, \nabla \phi_t)$ and defining the metric as a characterization of the tangent space

$$T_\rho^{WFR_\lambda} \mathcal{M}(\mathcal{X}) = \{ \dot{\rho} \mid \dot{\rho} = -\nabla \cdot (\rho \nabla \phi) + \lambda \rho \phi \} \tag{6}$$

$$\langle \dot{\mu}_t, \dot{\rho}_t \rangle_{T_\rho}^{WFR_\lambda} = \langle \phi_{\dot{\mu}_t}, \phi_{\dot{\rho}_t} \rangle_{T_\rho^*}^{WFR_\lambda} = \int \left( \langle \nabla \phi_{\dot{\mu}_t}, \nabla \phi_{\dot{\rho}_t} \rangle + \lambda \, \phi_{\dot{\mu}_t} \phi_{\dot{\rho}_t} \right) \rho \, dx. \tag{7}$$

## 2.3 ACTION MATCHING

Finally, Action Matching (AM) (Neklyudov et al., 2023) considers only the inner optimizations in Eq. (2) or Eq. (5) as a function of $v_t$ or $(v_t, g_t)$, assuming a distributional path $\mu_t$ is given via samples. In the $W_2$ case, to solve for the velocity $v_t = \nabla \phi_{\dot{\mu}_t}$ which corresponds to $\mu_t$ via the continuity equation or Eq. (3), Neklyudov et al. (2023) optimize the objective

$$\mathcal{A}[\mu_t] = \sup_{\phi_t} \int \phi_1 \mu_1 \, dx_1 - \int \phi_0 \mu_0 \, dx_0 - \int_0^1 \int \left( \frac{\partial \phi_t}{\partial t} + \frac{1}{2} \|\nabla \phi_t\|^2 \right) \mu_t \, dx_t dt, \tag{8}$$

over $\phi_t : \mathcal{X} \times [0, 1] \to \mathbb{R}$ parameterized by a neural network, with similar objectives for $WFR_\lambda$. To foreshadow our exposition in Sec. 3, we view Action Matching as maximizing a lower bound on the *action* $\mathcal{A}[\mu_t]$ or *kinetic energy* of the curve $\mu_t : [0, 1] \to \mathcal{P}_2(\mathcal{X})$ of densities. In particular, at the optimal $\phi_{\dot{\mu}_t}$ satisfying $\dot{\mu}_t = -\nabla \cdot (\mu_t \nabla \phi_{\dot{\mu}_t})$, the value of Eq. (8) becomes

$$\mathcal{A}[\mu_t] = \int_0^1 \frac{1}{2} \langle \dot{\mu}_t, \dot{\mu}_t \rangle_{T_{\mu_t}}^{W_2} dt = \int_0^1 \frac{1}{2} \langle \phi_{\dot{\mu}_t}, \phi_{\dot{\mu}_t} \rangle_{T_{\mu_t}^*}^{W_2} dt = \int_0^1 \int \frac{1}{2} \|\nabla \phi_t\|^2 \mu_t \, dx_t dt. \tag{9}$$

Our proposed framework for Wasserstein Lagrangian Flows considers minimizing the action functional over distributional paths, and our computational approach will include AM as a component.

## 3 WASSERSTEIN LAGRANGIAN FLOWS

In this section, we develop computational methods for optimizing Lagrangian action functionals on the space of (unnormalized) densities $\mathcal{P}(\mathcal{X})$.[1] Lagrangian actions are commonly used to define a cost function on the ground space $\mathcal{X}$, which is then 'lifted' to the space of densities via an optimal transport distance (Villani (2009) Ch. 7). We propose to formulate Lagrangians $\mathcal{L}[\rho_t, \dot{\rho}_t, t]$ *directly* in the density space, which includes OT with ground-space Lagrangian costs as a special case (App. B.1.2), but *also* allows us to consider kinetic and potential energy functionals which depend on the density and thus cannot be expressed using a ground-space Lagrangian. In particular, we consider kinetic energies capturing space-dependent birth-death terms (as in $WFR_\lambda$, Ex. 4.2) and potential energies capturing global information about the distribution of particles or cells (as in the SB problem, Ex. 4.4).

## 3.1 WASSERSTEIN LAGRANGIAN AND HAMILTONIAN FLOWS

We consider Lagrangian action functionals on the space of densities, defined in terms of a *kinetic energy* $\mathcal{K}[\rho_t, \dot{\rho}_t, t]$, which captures any dependence on the velocity of a curve $\dot{\rho}_t$, and a *potential energy* $\mathcal{U}[\rho_t, t]$ which depends only on the position $\rho_t$,

$$\mathcal{L}[\rho_t, \dot{\rho}_t, t] = \mathcal{K}[\rho_t, \dot{\rho}_t, t] - \mathcal{U}[\rho_t, t]. \tag{10}$$

---

[1]For convenience, we describe our methods using a generic $\mathcal{P}(\mathcal{X})$ (which may represent $\mathcal{P}_2(\mathcal{X})$ or $\mathcal{M}(\mathcal{X})$).

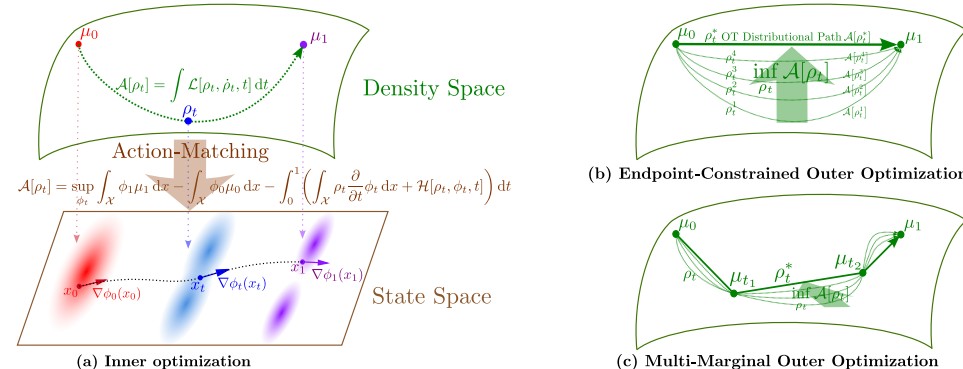

(a) Inner optimization

(b) Endpoint-Constrained Outer Optimization

(c) Multi-Marginal Outer Optimization

Figure 2: For different definitions of Lagrangian $\mathcal{L}[\rho_t, \dot{\rho}_t, t]$ or Hamiltonian $\mathcal{H}[\rho_t, s_t, t]$ on the space of densities, we obtain different action functionals $\mathcal{A}[\rho_t]$. Here, we show state-space velocity and optimal density paths for the $W_2$ geometry and OT problem. (a) The action functional for each curve can be evaluated using Action Matching (inner optimization in Thm. 1), which is performed in the state-space. (b,c) Minimization of the action functional (outer optimization in Thm. 1) is performed on the space of densities satisfying two endpoint constraints and possible intermediate constraints.

Throughout, we will assume $\mathcal{L}[\rho_t, \dot{\rho}_t, t]$ is lower semi-continuous (lsc) and strictly convex in $\dot{\rho}_t$.

Our goal is to solve for *Wasserstein Lagrangian Flows*, by optimizing the given Lagrangian over curves of densities $\rho_t : [0, 1] \rightarrow \mathcal{P}(\mathcal{X})$ which are constrained to pass through $M$ given points $\mu_{t_i} \in \mathcal{P}(\mathcal{X})$ at times $t_i$. We define the *action* of a curve $\mathcal{A}_{\mathcal{L}}[\rho_t] = \int_0^1 \mathcal{L}[\rho_t, \dot{\rho}_t, t]dt$ as the time-integral of the Lagrangian and seek the action-minimizing curve subject to the constraints

$$\mathcal{S}_{\mathcal{L}}(\{\mu_{t_i}\}_{i=0}^{M-1}) := \inf_{\rho_t \in \Gamma(\{\mu_{t_i}\})} \mathcal{A}_{\mathcal{L}}[\rho_t] := \inf_{\rho_t} \int_0^1 \mathcal{L}[\rho_t, \dot{\rho}_t, t]dt \quad \text{s.t.} \quad \rho_{t_i} = \mu_{t_i} \quad \forall \, 0 \le i \le M-1$$

(11)

where $\Gamma(\{\mu_{t_i}\}) = \{\rho_t : [0, 1] \rightarrow \mathcal{P}(\mathcal{X}) \mid \rho_0 = \mu_0, \, \rho_1 = \mu_1, \, \rho_{t_i} = \mu_{t_i} \quad (\forall \, 1 \le i \le M-2)\}$ indicates the set of curves matching the given constraints. We proceed writing the value as $\mathcal{S}$, leaving the dependence on $\mathcal{L}$ and $\mu_{t_i}$ implicit, but note $\mathcal{S}(\mu_{0,1})$ for $M = 2$ as an important special case.

Our objectives for solving Eq. (11) are based on the Hamiltonian $\mathcal{H}$ associated with the chosen Lagrangian. In particular, consider a cotangent vector $\phi_t \in T^*\mathcal{P}_2(\mathcal{X})$ or $\phi_t \in T^*\mathcal{M}(\mathcal{X})$, which is identified with a linear functional on the tangent space $\phi_t[\cdot] : \dot{\rho}_t \mapsto \int \phi_t \dot{\rho}_t dx_t$ via the canonical duality bracket. We define the *Hamiltonian* $\mathcal{H}[\rho_t, \phi_t, t]$ via the Legendre transform

$$\mathcal{H}[\rho_t, \phi_t, t] = \sup_{\dot{\rho}_t \in T_{\rho_t}\mathcal{P}} \int \phi_t \dot{\rho}_t \, dx_t - \mathcal{L}[\rho_t, \dot{\rho}_t, t] = \mathcal{K}^*[\rho_t, \phi_t, t] + \mathcal{U}[\rho_t, t],$$

(12)

where the sign of $\mathcal{U}[\rho_t, t]$ changes and $\mathcal{K}^*[\rho_t, \phi_t, t]$ translates the kinetic energy to the dual space. A primary example is when $\mathcal{K}[\rho_t, \dot{\rho}_t, t] = \frac{1}{2}\langle \dot{\rho}_t, \dot{\rho}_t \rangle_{T_{\rho_t}}$ is given by a Riemannian metric in the tangent space (such as for $W_2$ or $WFR_\lambda$), then $\mathcal{K}^*[\rho_t, \phi_t, t] = \frac{1}{2}\langle \phi_t, \phi_t \rangle_{T^*_{\rho_t}}$ is the same metric written in the cotangent space (see App. B.1 for detailed derivations for all examples considered in this work).

Finally, under our assumptions, $\mathcal{L}[\rho_t, \dot{\rho}_t, t]$ can also be written using the Legendre transform, $\mathcal{L}[\rho_t, \dot{\rho}_t, t] = \sup_{\phi_t \in T^*_{\rho_t}\mathcal{P}} \int \phi_t \dot{\rho}_t \, dx_t - \mathcal{H}[\rho_t, \phi_t, t]$. The following theorem forms the basis for our computational approach, and can be derived using the Legendre transform and integration by parts in time (see App. A for proof and Fig. 2 for visualization).

**Theorem 1.** *For a Lagrangian $\mathcal{L}[\rho_t, \dot{\rho}_t, t]$ which is lsc and strictly convex in $\dot{\rho}_t$, the optimization*

$$\mathcal{S} = \inf_{\rho_t \in \Gamma(\{\mu_{t_i}\})} \mathcal{A}_{\mathcal{L}}[\rho_t] = \inf_{\rho_t \in \Gamma(\{\mu_{t_i}\})} \int_0^1 \mathcal{L}[\rho_t, \dot{\rho}_t, t]dt$$

*is equivalent to the following dual*

$$\mathcal{S} = \inf_{\rho_t \in \Gamma(\{\mu_{t_i}\})} \sup_{\phi_t} \int \phi_1 \mu_1 \, dx_1 - \int \phi_0 \mu_0 \, dx_0 - \int_0^1 \left( \int \frac{\partial \phi_t}{\partial t} \rho_t dx_t + \mathcal{H}[\rho_t, \phi_t, t] \right) dt \quad (13)$$

*where, for $\phi_t \in T^*_{\rho_t}\mathcal{P}$, the Hamiltonian $\mathcal{H}[\rho_t, \phi_t, t]$ is the Legendre transform of $\mathcal{L}[\rho_t, \dot{\rho}_t, t]$ (Eq 12). In particular, the action $\mathcal{A}_{\mathcal{L}}[\rho_t]$ of a given curve is the solution to the inner optimization,*

$$\mathcal{A}_{\mathcal{L}}[\rho_t] = \sup_{\phi_t} \int \phi_1 \mu_1 \, dx_1 - \int \phi_0 \mu_0 \, dx_0 - \int_0^1 \left( \int \frac{\partial \phi_t}{\partial t} \rho_t dx_t + \mathcal{H}[\rho_t, \phi_t, t] \right) dt. \quad (14)$$

In line with our goal of defining Lagrangian actions *directly* on $\mathcal{P}(\mathcal{X})$ instead of via $\mathcal{X}$, Thm. 1 operates *only* in the abstract space of densities. See App. C for a detailed discussion.

Finally, the solution to the optimization in Eq. (11) can also be expressed using a Hamiltonian perspective, where the resulting *Wasserstein Hamiltonian flows* (Chow et al., 2020) satisfy the optimality conditions $\frac{\partial \rho_t}{\partial t} = \frac{\delta}{\delta \phi_t} \mathcal{H}[\rho_t, \phi_t, t]$ and $\frac{\partial \phi_t}{\partial t} = -\frac{\delta}{\delta \rho_t} \mathcal{H}[\rho_t, \phi_t, t]$.

To further analyze Thm. 1 and set the stage for our computational approach in Sec. 3.2, we consider the two optimizations in Eq. (13) as (i) *evaluating* the action functional $\mathcal{A}_{\mathcal{L}}[\rho_t]$ for a given curve $\rho_t$, and (ii) *optimizing* the action over curves $\rho_t \in \Gamma(\{\mu_{t_i}\})$ satisfying the desired constraints.

### 3.1.1 INNER OPTIMIZATION: EVALUATING $\mathcal{A}_{\mathcal{L}}[\rho_t]$ USING ACTION MATCHING

We immediately recognize the similarity of Eq. (14) to the AM objective in Eq. (8) for $\mathcal{H}[\rho_t, \phi_t, t] = \int \frac{1}{2}\|\nabla \phi_t\|^2 \rho_t dx_t$, which suggests a generalized notion of Action Matching as an inner loop to evaluate $\mathcal{A}_{\mathcal{L}}[\rho_t]$ for a given $\rho_t \in \Gamma(\{\mu_{t_i}\})$ in Thm. 1. For all $t$, the optimal cotangent vector $\phi_{\dot{\rho}_t}$ corresponds to the tangent vector $\dot{\rho}_t$ of the given curve via the Legendre transform or Eq. (14).

Neklyudov et al. (2023) assume access to samples from a *continuous* curve of densities $\mu_t$ which, from our perspective, corresponds to the limit as the number of constraints $M \to \infty$. Since $\rho_t \in \Gamma(\{\mu_{t_i}\})$ has no remaining degrees of freedom in this case, the outer optimization over $\rho_t$ can be ignored and expectations in Eq. (8) are written directly under $\mu_t$. However, this assumption is often unreasonable in applications such as trajectory inference, where data is sampled discretely in time.

### 3.1.2 OUTER OPTIMIZATION OVER CONSTRAINED DISTRIBUTIONAL PATHS

In our settings of interest, the outer optimization over curves $\mathcal{S}_{\mathcal{L}}(\{\mu_{t_i}\}) = \inf_{\rho_t \in \Gamma(\{\mu_{t_i}\})} \mathcal{A}_{\mathcal{L}}[\rho_t]$ is thus necessary to *interpolate* between $M$ given marginals using the inductive bias encoded in the Lagrangian $\mathcal{L}[\rho_t, \dot{\rho}_t, t]$. Crucially, our parameterization of $\rho_t$ in Sec. 3.2.2 will enforce $\rho_t \in \Gamma(\{\mu_{t_i}\})$ by design, given access to samples from $\mu_{t_i}$. Nevertheless, upon reaching an optimal $\rho_t$, our primary object of interest is the dynamics model corresponding to $\dot{\rho}_t$ and parameterized by the optimal $\phi_{\dot{\rho}_t}$ in Eq. (14), which may be used to transport particles or predict individual trajectories.

## 3.2 COMPUTATIONAL APPROACH FOR SOLVING WASSERSTEIN LAGRANGIAN FLOWS

In this section, we describe our computational approach to solving for a class of Wasserstein Lagrangian Flows, which is summarized in Alg. 1.

### 3.2.1 LINEARIZABLE KINETIC AND POTENTIAL ENERGIES

Despite the generality of Thm. 1, we restrict attention to Lagrangians with the following property.

**Definition 3.1** ((Dual) Linearizability). *A Lagrangian $\mathcal{L}[\rho_t, \dot{\rho}_t, t]$ is dual linearizable if the corresponding Hamiltonian $\mathcal{H}[\rho_t, \phi_t, t]$ can be written as a linear functional of the density $\rho_t$. In other words, $\mathcal{H}[\rho_t, \phi_t, t]$ is* linearizable *if there exist functions $K^*(x_t, \phi_t, t)$, and $U(x_t, \phi_t, t)$ such that*

$$\mathcal{H}[\rho_t, \phi_t, t] = \int \Big( K^*(x_t, \phi_t, t) + U(x_t, \phi_t, t) \Big) \rho_t dx_t. \tag{15}$$

This property suggests that we only need to draw samples from $\rho_t$ and need not evaluate its density, which allows us to derive an efficient parameterization of curves satisfying $\rho_t \in \Gamma(\{\mu_{t_i}\})$ below. [2]

As examples, note that the $WFR_\lambda$ or $W_2$ metrics as the Lagrangian yield a linear Hamiltonian $\mathcal{H}[\rho_t, \phi_t, t] = \mathcal{K}^*[\rho_t, \phi_t, t] = \frac{1}{2}\langle \phi_t, \phi_t \rangle_{T^*_{\rho_t}} = \int (\frac{1}{2}\|\nabla \phi_t\|^2 + \frac{\lambda}{2}\phi_t^2)\rho_t dx_t$, with $\lambda = 0$ for $W_2$. Potential energies $\mathcal{U}[\rho_t, t] = \int V_t(x_t)\rho_t dx_t$ which are linear in $\rho_t$ (Ex. 4.3) clearly satisfy Def. 3.1. However, nonlinear potential energies as in Ex. 4.4 require reparameterization to be linearizable.

### 3.2.2 PARAMETERIZATION AND OPTIMIZATION

For any Lagrangian optimization with a linearizable dual objective as in Def. 3.1, we consider parameterizing the cotangent vectors $\phi_t$ and the distributional path $\rho_t$. We parameterize $\phi_t$ as a neural network $\phi_t(x, \theta)$ which takes $t$ and $x$ as inputs with parameters $\theta$, and outputs a scalar. Inspired

---

[2]In App. B.3, we highlight the Schrödinger Equation as a special case of our framework which does not appear to admit a linear dual problem. In this case, optimization of Eq. (13) may require explicit modeling of the density $\rho_t$ corresponding to a given set of particles $x_t$ (e.g. see Pfau et al. (2020)).

---

**Algorithm 1** Learning Wasserstein Lagrangian Flows

---

**Require:** samples from the marginals $\mu_0, \mu_1$, parametric model $\phi_t(x, \theta)$, generator from $\rho_t(x, \eta)$

  **for** learning iterations **do**

    sample from marginals $\{x_0^i\}_{i=1}^n \sim \mu_0$, $\{x_1^i\}_{i=1}^n \sim \mu_1$, sample time $\{t^i\}_{i=1}^n \sim \text{UNIFORM}[0, 1]$

    $x_t^i = (1 - t^i)x_0^i + t^i x_1^i + t^i(1 - t^i)\text{NNET}(t^i, x_0^i, x_1^i; \eta)$

    $-\text{GRAD}_\eta = \nabla_\eta \frac{1}{n} \sum_i^n \left[ \frac{\partial \phi_{t^i}}{\partial t^i}(x_t^i(\eta), \theta) + K^*\big(x, \phi_{t^i}(x_t^i(\eta), \theta), t^i\big) + U\big(x, \phi_{t^i}(x_t^i(\eta), \theta), t^i\big) \right]$

    **for** Wasserstein gradient steps **do**

      $x_t^i \leftarrow x_t^i + \alpha \cdot t^i(1 - t^i)\nabla_x \left[ \frac{\partial \phi_{t^i}}{\partial t^i}(x_t^i, \theta) + K^*\big(x, \phi_{t^i}(x_t^i, \theta), t^i\big) + U\big(x, \phi_{t^i}(x_t^i, \theta), t^i\big) \right]$

    **end for**

    $\text{GRAD}_\theta = \nabla_\theta \frac{1}{n} \sum_i^n \left[ \phi_1(x_1^i, \theta) - \phi_0(x_0^i, \theta) - \frac{\partial \phi_{t^i}(x_t^i, \theta)}{\partial t^i} - K^*\big(x, \phi_{t^i}(x_t^i, \theta), t^i\big) - U\big(x, \phi_{t^i}(x_t^i, \theta), t^i\big) \right]$

    update parameters using gradients $\text{GRAD}_\eta, \text{GRAD}_\theta$

  **end for**

  **return** cotangent vectors $\phi_t(x, \theta)$

---

by the fact that we only need to draw samples from $\rho_t$ for these problems, we parameterize the distribution path $\rho_t(x, \eta)$ as a generative model, where the samples are generated as follows

$$x_t = (1 - t)x_0 + tx_1 + t(1 - t)\text{NNET}(t, x_0, x_1; \eta), \quad x_0 \sim \mu_0, \quad x_1 \sim \mu_1. \tag{16}$$

Notably, this preserves the endpoint marginals $\mu_0, \mu_1$. For multiple constraints, we can modify our sampling procedure to interpolate between two intermediate dataset marginals, with neural network parameters $\eta$ shared across timesteps

$$x_t = \frac{t_{i+1} - t}{t_{i+1} - t_i}x_{t_i} + \frac{t - t_i}{t_{i+1} - t_i}x_{t_{i+1}} + \left( 1 - \left( \frac{t_{i+1} - t}{t_{i+1} - t_i} \right)^2 - \left( \frac{t - t_i}{t_{i+1} - t_i} \right)^2 \right)\text{NNET}(t, x_{t_i}, x_{t_{i+1}}; \eta) \, .$$

For linearizable dual objectives as in Eq. (13) and Eq. (15), we optimize

$$\text{LOSS}(\theta, \eta) = \min_\eta \max_\theta \int \phi_1(x_1, \theta)\mu_1 \, dx_1 - \int \phi_0(x_0, \theta)\mu_0 \, dx_0 \tag{17}$$

$$- \int_0^1 \int \left( \frac{\partial \phi_t}{\partial t}(x_t, \theta) + K^*\big(x_t, \phi_t(x_t, \theta), t\big) + U\big(x_t, \phi_t(x_t, \theta), t\big) \right)\rho_t(x_t, \eta)dx_t dt,$$

where the optimization w.r.t. $\eta$ is performed via the re-parameterization trick. An alternative to parametrizing the distributional path $\rho_t$ is to perform minimization of Eq. (17) via the Wasserstein gradient flow, i.e. the samples $x_t$ from the initial path $\rho_t$ are updated as follows

$$x_t' = x_t + \alpha \cdot t(1 - t)\nabla_x \left[ \frac{\partial \phi_t}{\partial t}(x_t, \theta) + K^*\big(x, \phi_t(x_t, \theta), t\big) + U\big(x, \phi_t(x_t, \theta), t\big) \right], \tag{18}$$

where $\alpha$ is a hyperparameter regulating the step-size, and the coefficient $t(1 - t)$ guarantees the preservation of the endpoints. In practice, we find that combining both the parametric and nonparametric approaches works best. The pseudo-code for the resulting algorithm is given in Alg. 1.

**Discontinuous Interpolation** The support of the optimal distribution path $\rho_t$ might be disconnected, as in Fig. 1a. Thus, it may be impossible to interpolate continuously between independent samples from the marginals while staying in the support of the optimal path. To allow for a discontinuous interpolation, we pass a discontinuous indicator variable $\mathbb{1}[t < 0.5]$ to the model $\rho_t(x, \eta)$. This indicator is crucial to ensure our parameterization is expressive enough to approximate any suitable distributional path including, for example, the optimal OT path (see proof in App. D).

**Proposition 2.** *For any absolutely-continuous distributional path $\rho_t : [0, 1] \mapsto \mathcal{P}_2(\mathcal{X})$ on the $W_2$ manifold, there exists a function $\text{NNET}^*(t, x_0, x_1, \mathbb{1}[t < 0.5]; \eta)$ such that Eq. (16) samples from $\rho_t$.*

## 4 EXAMPLES OF WASSERSTEIN LAGRANGIAN FLOWS

We now analyze the Lagrangians, dual objectives, and Hamiltonian optimality conditions corresponding to several important examples of Wasserstein Lagrangian flows. We present various kinetic and potential energy terms using their motivating examples and $M = 2$ endpoint constraints. However, note that we may combine various energy terms to construct Lagrangians $\mathcal{L}[\rho_t, \dot{\rho}_t, t]$, and optimize subject to multiple constraints, as we consider in our experiments in Sec. 5.

**Example 4.1** ($W_2$ **Optimal Transport**)**.** The Benamou-Brenier formulation of $W_2$ optimal transport in Eq. (2) is the simplest example of our framework, with no potential energy and the kinetic energy defined by the Otto metric $\mathcal{L}[\rho_t, \dot{\rho}_t, t] = \frac{1}{2}\langle \dot{\rho}_t, \dot{\rho}_t \rangle_{T_{\rho_t}}^{W_2} = \mathcal{H}[\rho_t, \phi_{\dot{\rho}_t}, t] = \frac{1}{2}\int \|\nabla \phi_{\dot{\rho}_t}\|^2 \rho_t dx_t.$

In contrast to Eq. (2), note that our Lagrangian optimization in Eq. (11) is over $\rho_t$ only, while solving the dual objective introduces the second optimization to identify $\phi_{\dot\rho_t}$ such that $\dot\rho_t = -\nabla \cdot (\rho_t \nabla \phi_{\dot\rho_t})$. Our dual objective for solving the standard optimal transport problem with quadratic cost becomes

$$\mathcal{S}_{OT} = \inf_{\rho_t \in \Gamma(\mu_0,\mu_1)} \sup_{\phi_t} \int \phi_1 d\mu_1 - \int \phi_0 d\mu_0 - \int_0^1 \int \left( \frac{\partial \phi_t}{\partial t} + \frac{1}{2}\|\nabla\phi_t\|^2 \right) \rho_t dx_t dt, \qquad (19)$$

where the Hamiltonian optimality conditions $\frac{\partial \rho_t}{\partial t} = \frac{\delta}{\delta \phi_t}\mathcal{H}[\rho_t,\phi_t,t]$ and $\frac{\partial \phi_t}{\partial t} = -\frac{\delta}{\delta \rho_t}\mathcal{H}[\rho_t,\phi_t,t]$ (Chow et al., 2020) recover the characterization of the Wasserstein geodesics via the continuity and Hamilton-Jacobi equations (Benamou & Brenier, 2000),

$$\dot\rho_t = -\nabla \cdot (\rho_t \nabla \phi_t) \qquad \frac{\partial \phi_t}{\partial t} + \frac{1}{2}\|\nabla\phi_t\|^2 = 0. \qquad (20)$$

It is well known that optimal transport plans (or Wasserstein-2 geodesics) are 'straight-paths' in the Euclidean space (Villani, 2009). For the flow induced by a vector field $\nabla\phi_t$, we calculate the acceleration, or second derivative with respect to time, as

$$\ddot{X}_t = \nabla \left[ \frac{\partial \phi_t}{\partial t} + \frac{1}{2}\|\nabla\phi_t\|^2 \right] = 0, \qquad (21)$$

where zero acceleration is achieved if $\frac{\partial \phi_t}{\partial t} + \frac{1}{2}\|\nabla\phi_t\|^2 = c, \forall t$, as occurs at optimality in Eq. (20).

**Example 4.2** (**Unbalanced Optimal Transport**). The *unbalanced* OT problem arises from the $WFR_\lambda$ geometry, and is useful for modeling mass teleportation and changes in total probability mass when cell birth and death occur as part of the underlying dynamics (Schiebinger et al., 2019; Lu et al., 2019). Viewing the dynamical formulation of WFR in Eq. (5) as a Lagrangian optimization,

$$WFR_\lambda(\mu_0,\mu_1)^2 = \inf_{\rho_t \in \Gamma(\mu_0,\mu_1)} \int_0^1 \mathcal{L}[\rho_t,\dot\rho_t,t]dt = \int_0^1 \frac{1}{2}\langle \dot\rho_t,\dot\rho_t \rangle_{T_{\rho_t}}^{WFR_\lambda} dt \text{ s.t. } \rho_0 = \mu_0, \rho_1 = \mu_1.$$

Compared to Eq. (5), our Lagrangian formulation again optimizes over $\rho_t$ only, and solving the dual requires finding $\phi_{\dot\rho_t}$ such that $\dot\rho_t = -\nabla \cdot (\rho_t \nabla \phi_{\dot\rho_t}) + \lambda \rho_t \phi_{\dot\rho_t}$ as in Eq. (6). We optimize the objective

$$\mathcal{S}_{uOT} = \inf_{\rho_t \in \Gamma(\mu_0,\mu_1)} \sup_{\phi_t} \int \phi_1 d\mu_1 - \int \phi_0 d\mu_0 - \int_0^1 \int \left( \frac{\partial \phi_t}{\partial t} + \frac{1}{2}\|\nabla\phi_t\|^2 + \frac{\lambda}{2}\phi_t^2 \right) \rho_t dx_t dt,$$

where we recognize the $WFR_\lambda$ cotangent metric from Eq. (7) in the final term, $\mathcal{H}[\rho_t,\phi_t,t] = \mathcal{K}^*[\rho_t,\phi_t,t] = \frac{1}{2}\langle \phi_t,\phi_t \rangle_{T_{\rho_t}^*}^{WFR_\lambda} = \frac{1}{2}\int \left( \|\nabla\phi_t\|^2 + \lambda\,\phi_t^2 \right) \rho_t\,dx_t$.

**Example 4.3** (**Physically Constrained Optimal Transport**). A popular technique for incorporating inductive bias from biological or geometric prior information into trajectory inference methods is to consider spatial potentials $\mathcal{U}[\rho_t,t] = \int V_t(x_t)\rho_t dx_t$ (Tong et al., 2020; Koshizuka & Sato, 2022; Pooladian et al., 2023b), which are already linear in the density. In this case, we may consider *any* linearizable kinetic energy (see App. B.1). For the $W_2$ transport case, our objective is

$$\mathcal{S}_{pOT} = \inf_{\rho_t \in \Gamma(\mu_0,\mu_1)} \sup_{\phi_t} \int \phi_1 d\mu_1 - \int \phi_0 d\mu_0 - \int_0^1 \int \left( \frac{\partial \phi_t(x_t)}{\partial t} + \frac{1}{2}\|\nabla\phi_t(x_t)\|^2 + V_t(x_t) \right) \rho_t dx_t dt,$$

with the optimality conditions

$$\dot\rho_t = -\nabla \cdot (\rho_t \nabla \phi_t), \qquad \frac{\partial \phi_t}{\partial t} + \frac{1}{2}\|\nabla\phi_t\|^2 + V_t = 0, \qquad \ddot{X}_t = \nabla \left[ \frac{\partial \phi_t}{\partial t} + \frac{1}{2}\|\nabla\phi_t\|^2 \right] = -\nabla V_t. \quad (22)$$

As in Eq. (21), the latter condition implies that the acceleration is given by the gradient of the spatial potential $V_t(x_t)$. We describe the potentials used in our experiments on scRNA datasets in Sec. 5.

**Example 4.4** (**Schrödinger Bridge**). For many problems of interest, such as scRNA sequencing (Schiebinger et al., 2019), it may be useful to incorporate stochasticity into the dynamics as prior knowledge. For Brownian-motion diffusion processes with known coefficient $\sigma$, the dynamical Schrödinger Bridge (SB) problem (Mikami, 2008; Léonard, 2013; Chen et al., 2021b) is given by

$$\mathcal{S}_{SB} = \inf_{\rho_t,v_t} \int_0^1 \int \frac{1}{2}\|v_t\|^2 \rho_t dx_t dt \quad \text{s.t. } \dot\rho_t = -\nabla \cdot (\rho_t v_t) + \frac{\sigma^2}{2}\Delta\rho_t, \ \rho_0 = \mu_0, \ \rho_1 = \mu_1. \qquad (23)$$

To model the SB problem, we consider the following potential energy with the $W_2$ kinetic energy,

$$\mathcal{U}[\rho_t,t] = -\frac{\sigma^4}{8}\int \|\nabla \log \rho_t\|^2 \rho_t dx_t, \qquad (24)$$

Table 1: Results for high-dim PCA representation of single-cell data for corresponding datasets. We report Wasserstein-1 distance averaged over left-out marginals. All results are averaged over 5 independent runs. Results with citations are taken from corresponding papers.

| Method | dim=5 EB | dim=50 Cite | Multi | dim=100 Cite | Multi |
|---|---|---|---|---|---|
| exact OT | 0.822 | 37.569 | 47.084 | 42.974 | 53.271 |
| WLF-OT (ours) | $0.814 \pm 0.002$ | $38.253 \pm 0.071$ | $47.736 \pm 0.110$ | $44.769 \pm 0.054$ | $55.313 \pm 0.754$ |
| OT-CFM (more parameters) | $0.822 \pm 3.0\text{e-}4$ | $37.821 \pm 0.010$ | $47.268 \pm 0.017$ | $44.013 \pm 0.010$ | $54.253 \pm 0.012$ |
| OT-CFM (Tong et al., 2023b) | $\mathbf{0.790} \pm 0.068$ | $38.756 \pm 0.398$ | $47.576 \pm 6.622$ | $45.393 \pm 0.416$ | $54.814 \pm 5.858$ |
| I-CFM (Tong et al., 2023b) | $0.872 \pm 0.087$ | $41.834 \pm 3.284$ | $49.779 \pm 4.430$ | $48.276 \pm 3.281$ | $57.262 \pm 3.855$ |
| WLF-UOT ($\lambda = 1$, ours) | $\mathbf{0.800} \pm 0.002$ | $\mathbf{37.035} \pm 0.079$ | $45.903 \pm 0.161$ | $\mathbf{43.530} \pm 0.067$ | $53.403 \pm 0.168$ |
| WLF-SB (ours) | $0.816 \pm 7.7\text{e-}4$ | $39.240 \pm 0.068$ | $47.788 \pm 0.111$ | $46.177 \pm 0.083$ | $55.716 \pm 0.058$ |
| $[\text{SF}]^2$ M-Geo (Tong et al., 2023a) | $1.221 \pm 0.38$ | $38.524 \pm 0.293$ | $\mathbf{44.795} \pm 1.911$ | $44.498 \pm 0.416$ | $\mathbf{52.203} \pm 1.957$ |
| $[\text{SF}]^2$ M-Exact (Tong et al., 2023a) | $\mathbf{0.793} \pm 0.066$ | $40.009 \pm 0.783$ | $45.337 \pm 2.833$ | $46.530 \pm 0.426$ | $52.888 \pm 1.986$ |
| WLF-(OT + potential, ours) | $0.651 \pm 0.002$ | $36.167 \pm 0.031$ | $38.743 \pm 0.060$ | $42.857 \pm 0.045$ | $47.365 \pm 0.051$ |
| WLF-(UOT + potential, $\lambda = 1$, ours) | $\mathbf{0.634} \pm 0.001$ | $\mathbf{34.160} \pm 0.041$ | $\mathbf{36.131} \pm 0.023$ | $\mathbf{41.084} \pm 0.043$ | $\mathbf{45.231} \pm 0.010$ |

which arises from the entropy $\mathcal{F}[\rho_t] = -H[\rho_t] = \int (\log \rho_t - 1) \rho_t dx_t$ via $\nabla \frac{\delta}{\delta \rho_t} \mathcal{F}[\rho_t] = \nabla \log \rho_t$. We assume time-independent $\sigma$ to simplify $\mathcal{U}[\rho_t, t]$, but consider time-varying $\sigma_t$ in Ex. B.2.

To transform the potential energy term into a dual-linearizable form for the SB problem, we consider the reparameterization $\Phi_t = \phi_t + \frac{\sigma^2}{2} \log \rho_t$, which translates between the drift $\nabla \phi_t$ of the probability flow ODE and the drift $\nabla \Phi_t$ of the Fokker-Planck equation (Song et al., 2020). With detailed derivations in App. B.2, the dual objective becomes

$$\mathcal{S}_{SB} = \inf_{\rho_t \in \Gamma(\mu_0, \mu_1)} \sup_{\Phi_t} \int \Phi_1 d\mu_1 - \int \Phi_0 d\mu_0 - \int_0^1 \int \left( \frac{\partial \Phi_t}{\partial t} + \frac{1}{2} \|\nabla \Phi_t\|^2 + \frac{\sigma^2}{2} \Delta \Phi_t \right) \rho_t dx_t dt. \quad (25)$$

## 5 EXPERIMENTS

We apply our methods for trajectory inference of single-cell RNA sequencing data, including the Embryoid body (**EB**) dataset (Moon et al., 2019), CITE-seq (**Cite**) and Multiome (**Multi**) datasets (Burkhardt et al., 2022), and melanoma treatment dataset of (Bunne et al., 2021; Pariset et al., 2023).

**Potential for Physically-Constrained OT**  For all tasks, we consider the simplest possible model of the physical potential accelerating the cells. For each marginal except the first and the last ones, we estimate the acceleration of its mean using finite differences. The potential for the corresponding time interval is then $V_t(x) = -\langle x, a_t \rangle$, where $a_t$ is the estimated acceleration of the mean value. For leave-one-out tasks, we include the mean of the left out marginal since the considered data contains too few marginals (4 for Cite and Multi) for learning a meaningful model of the acceleration.

**Leave-One-Out Marginal Task**  To test the ability of our approaches to approximate interpolating marginal distributions, we follow Tong et al. (2020) and evaluate models using a leave-one-timepoint-out strategy. In particular, we train on all marginals except at time $t_i$, and evaluate by computing the Wasserstein-1 distance between the predicted marginal $\rho_{t_i}$ and the left-out marginal $\mu_{t_i}$. For preprocessing and baselines, we follow Tong et al. (2023a;b) (see App. E.1 for details).

In Table 1, we report results on EB, Cite, and Multi datasets. First, we see that our proposed WLF-OT method achieves comparable results to related approaches: OT-CFM and I-CFM (Tong et al., 2023b), which use minibatch OT couplings or independent samples of the marginals, respectively. For OT-CFM, we reproduce the results using a larger model to match the performance of the exact OT solver (Flamary et al., 2021). These models represent dynamics with minimal prior knowledge, and thus serve as a baseline when compared against dynamics incorporating additional priors.

Next, we consider Lagrangians encoding various prior information. WLF-SB (ours), $[\text{SF}]^2$ M-Exact (Tong et al., 2023a), and SB-CFM (Tong et al., 2023b) incorporate stochasticity into the dynamics by solving the SB problem; $[\text{SF}]^2$ M-Geo takes advantage of the data manifold geometry by learning from OT couplings generated with the approximate geodesic cost; our WLF-UOT incorporates probability mass teleportation using the $WFR$ kinetic energy. In Table 1, we see that WLF-UOT yields consistent performance improvements across datasets. Finally, we observe that a good model of the potential function can drastically improve performance, using either $W_2$ or $WFR$ kinetic energy.

**Comparison with SB Baselines on EB Dataset**  To compare against a broader class of baselines for the SB problem, we consider the setting of Koshizuka & Sato (2022, Table 1) on the EB dataset. Instead of leaving out one marginal, we divide the data using a train/test split and evaluate the W1 distance between the test $\mu_{t_i}$ and $\rho_{t_i}$ obtained by running dynamics from the previous $\mu_{t_{i-1}}$. In Table 2, we find that WLF-SB outperforms several SB baselines from recent literature (see Sec. 6).

Table 2: Results for train/test splits of 5-dim PCA on EB dataset, with the setting and baseline results taken from Koshizuka & Sato (2022, Table 1). We report W1 distance between test $\mu_{t_i}$ and $\rho_{t_i}$ obtained by running dynamics from $\mu_{t_{i-1}}$.

| Model | $t_1$ | $t_2$ | $t_3$ | $t_4$ | Mean |
|---|---|---|---|---|---|
| Neural SDE (Li et al., 2020) | 0.69 | 0.91 | 0.85 | 0.81 | 0.82 |
| TrajectoryNet (Tong et al., 2020) | 0.73 | 1.06 | 0.90 | 1.01 | 0.93 |
| IPF (GP) (Vargas et al., 2021) | 0.70 | 1.04 | 0.94 | 0.98 | 0.92 |
| IPF (NN) (De Bortoli et al., 2021) | 0.73 | 0.89 | 0.84 | 0.83 | 0.82 |
| SB-FBSDE (Chen et al., 2021a) | 0.56 | 0.80 | 1.00 | 1.00 | 0.84 |
| NLSB (Koshizuka & Sato, 2022) | 0.68 | 0.84 | 0.81 | 0.79 | 0.78 |
| WLF-OT | 0.65 | 0.78 | 0.76 | 0.75 | 0.74 |
| WLF-SB | 0.63 | 0.79 | 0.77 | 0.74 | **0.73** |
| WLF-(OT + potential) | 0.64 | 0.77 | 0.76 | 0.76 | **0.73** |
| WLF-UOT ($\lambda = 0.1$) | 0.64 | 0.84 | 0.80 | 0.81 | 0.77 |
| WLF-(UOT + potential, $\lambda = 0.1$) | 0.67 | 0.80 | 0.78 | 0.78 | 0.76 |

Table 3: Results in the setting of Pariset et al. (2023, Table 1) (uDSB) for melanoma treatment data with 3 marginals and train/test splits. We report test MMD and W2 distance between $\mu_1$ and $\rho_1$ obtained by running dynamics from $\mu_0$.

| Model | MMD | $W_2$ |
|---|---|---|
| SB-FBSDE (Chen et al., 2021a) | 1.86e-2 | 6.23 |
| uDSB (no growth) (Pariset et al., 2023) | 1.86e-2 | 6.27 |
| uDSB (w/growth) (Pariset et al., 2023) | 1.75e-2 | 6.11 |
| WLF-OT (no growth) | **5.04e-3** | 5.20 |
| WLF-UOT ($\lambda = 0.1$) | 9.16e-3 | **5.01** |

**Comparison with UOT Baseline on Melanoma Dataset** To test the ability of our WLF-OT approach to account for cell birth and death, we consider the 50-dim. setting of Pariset et al. (2023, Table 1) for melanoma cells undergoing treatment with a cancer drug. In Table 3, we show that WLF-OT and WLF-UOT can outperform the unbalanced baseline (uDSB) from Pariset et al. (2023).

## 6 RELATED WORK

**Wasserstein Hamiltonian Flows** Chow et al. (2020) develop the notion of a Hamiltonian flow on the Wasserstein manifold and consider several of the same examples discussed here. While the Hamiltonian and Lagrangian formalisms describe the same integral flow through optimality conditions for $(\rho_t, \dot{\rho}_t)$ and $(\rho_t, \phi_t)$, Chow et al. (2020); Wu et al. (2023) emphasize solving the Cauchy problem suggested by the Hamiltonian perspective. Our approach recovers the Hamiltonian flow $(\rho_t, \phi_t)$ in the cotangent bundle at optimality, but does so by solving a variational problem.

**Flow Matching and Diffusion Schrödinger Bridge Methods** Flow Matching methods (Liu, 2022; Lipman et al., 2022; Albergo & Vanden-Eijnden, 2022; Albergo et al., 2023; Tong et al., 2023b;a) learn a marginal vector field corresponding to a mixture-of-bridges process parameterized by a coupling and interpolating bridge (Shi et al., 2023). When samples from the endpoint marginals are coupled via an OT plan, Flow Matching solves a dynamical optimal transport problem (Pooladian et al., 2023a). Rectified Flow obtains couplings using ODE simulation with the goal of straight-path trajectories for generative modeling (Liu, 2022; Liu et al., 2022b), which is extended to SDEs in bridge matching methods (Shi et al., 2023; Peluchetti, 2023). Diffusion Schrödinger Bridge (DSB) methods (De Bortoli et al., 2021; Chen et al., 2021a) also update the couplings iteratively based on learned forward and backward SDEs, and have recently been adapted to solve the unbalanced OT problem in Pariset et al. (2023). Finally, Liu et al. (2022a; 2023) consider extending DSB or bridge matching methods to solve physically-constrained SB problems. Unlike the above methods, our approach does not require optimal couplings to sample from the intermediate marginals, and thus avoids both simulating ODEs or SDEs and running minibatch (regularized) OT solvers.

**Optimal Transport with Lagrangian Cost** Input-convex neural networks (Amos et al., 2017) provide an efficient approach to static OT (Makkuva et al., 2020; Korotin et al., 2021; Bunne et al., 2021; 2022) but are limited to the Euclidean cost. Several works extend to other costs using static (Fan et al., 2022; Pooladian et al., 2023b; Uscidda & Cuturi, 2023) or dynamical formulations (Liu et al., 2021; Koshizuka & Sato, 2022). The most general way to define a transport cost is via a Lagrangian action in the state-space (Villani (2009) Ch. 7). While we focus on lifted Lagrangians in the density space, our framework encompasses OT with state-space Lagrangian costs (App. B.1.2).

## 7 CONCLUSION

In this work, we demonstrated that many variations of optimal transport, such as Schrödinger bridge, unbalanced OT, or OT with physical constraints can be formulated as Lagrangian action minimization on the density manifold. We proposed a computational framework for this minimization by deriving a dual objective in terms of cotangent vectors, which correspond to a vector field on the state-space and can be parameterized via a neural network. As an application, we studied the problem of trajectory inference in biological systems, and showed that we can incorporate prior knowledge of the dynamics while respecting marginal constraints on the observed data, resulting in significant improvement in several benchmarks. We expect our approach can be extended to other natural science domains such as quantum mechanics and social sciences by incorporating new prior information for learning the underlying dynamics.

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

## A    GENERAL DUAL OBJECTIVES FOR WASSERSTEIN LAGRANGIAN FLOWS

In this section, we derive the general forms for the Hamiltonian dual objectives arising from Wasserstein Lagrangian Flows. We prove Thm. 1 and derive the general dual objective in Eq. (13) of the main text, before considering the effect of multiple marginal constraints in App. A.1. We defer explicit calculation of Hamiltonians for important special cases to App. B.

**Theorem 1.** *For a Lagrangian $\mathcal{L}[\rho_t, \dot\rho_t, t]$ which is lsc and strictly convex in $\dot\rho_t$, the optimization*

$$\mathcal{S} = \inf_{\rho_t \in \Gamma(\{\mu_{t_i}\})} \mathcal{A}_{\mathcal{L}}[\rho_t] = \inf_{\rho_t \in \Gamma(\{\mu_{t_i}\})} \int_0^1 \mathcal{L}[\rho_t, \dot\rho_t, t] dt$$

*is equivalent to the following dual*

$$\mathcal{S} = \inf_{\rho_t \in \Gamma(\{\mu_{t_i}\})} \sup_{\phi_t} \int \phi_1 \mu_1 \, dx_1 - \int \phi_0 \mu_0 \, dx_0 - \int_0^1 \left( \int \frac{\partial \phi_t}{\partial t} \rho_t dx_t + \mathcal{H}[\rho_t, \phi_t, t] \right) dt \quad (13)$$

*where, for $\phi_t \in T_{\rho_t}^* \mathcal{P}$, the Hamiltonian $\mathcal{H}[\rho_t, \phi_t, t]$ is the Legendre transform of $\mathcal{L}[\rho_t, \dot\rho_t, t]$ (Eq 12). In particular, the action $\mathcal{A}_{\mathcal{L}}[\rho_t]$ of a given curve is the solution to the inner optimization,*

$$\mathcal{A}_{\mathcal{L}}[\rho_t] = \sup_{\phi_t} \int \phi_1 \mu_1 \, dx_1 - \int \phi_0 \mu_0 \, dx_0 - \int_0^1 \left( \int \frac{\partial \phi_t}{\partial t} \rho_t dx_t + \mathcal{H}[\rho_t, \phi_t, t] \right) dt. \quad (14)$$

Recall the definition of the Legendre transform for $\mathcal{L}[\rho_t, \dot\rho_t, t]$ strictly convex in $\dot\rho_t$,

$$\mathcal{H}[\rho_t, \phi_t, t] = \sup_{\dot\rho_t \in \mathcal{T}_{\rho_t} \mathcal{P}} \int \phi_t \dot\rho_t \, dx_t - \mathcal{L}[\rho_t, \dot\rho_t, t] \quad (26)$$

$$\mathcal{L}[\rho_t, \dot\rho_t, t] = \sup_{\phi_t \in \mathcal{T}_{\rho_t}^* \mathcal{P}} \int \phi_t \dot\rho_t \, dx_t - \mathcal{H}[\rho_t, \phi_t, t] \quad (27)$$

*Proof.* We prove the case of $M = 2$ here and the case of $M > 2$ below in App. A.1.

Denote the set of curves of marginal densities $\rho_t$ with the prescribed endpoint marginals as $\Gamma(\mu_0, \mu_1) = \{\rho_t | \rho_t \in \mathcal{P}(\mathcal{X}) \, \forall t, \rho_0 = \mu_0, \rho_1 = \mu_1\}$. The result follows directly from the definition of the Legendre transform in Eq. (26) and integration by parts in time in step $(i)$,

$$\mathcal{S}_{\mathcal{L}}(\{\mu_{0,1}\}) = \inf_{\rho_t} \int_0^1 \mathcal{L}[\rho_t, \dot\rho_t, t] dt \quad \text{s.t.} \quad \rho_0 = \mu_0, \qquad \rho_1 = \mu_1 \quad (28)$$

$$= \inf_{\rho_t \in \Gamma(\mu_0, \mu_1)} \int_0^1 \mathcal{L}[\rho_t, \dot\rho_t, t] dt$$

$$= \inf_{\rho_t \in \Gamma(\mu_0, \mu_1)} \sup_{\phi_t \in \mathcal{T}_{\rho_t}^* \mathcal{P}} \int_0^1 \left( \int \phi_t \dot\rho_t \, dx_t - \mathcal{H}[\rho_t, \phi_t, t] \right) dt$$

$$\overset{(i)}{=} \inf_{\rho_t \in \Gamma(\mu_0, \mu_1)} \sup_{\phi_t} \int \phi_1 \rho_1 dx_1 - \int \phi_0 \rho_0 dx_0 - \int_0^1 \left( \int \frac{\partial \phi_t}{\partial t} \rho_t \, dx_t + \mathcal{H}[\rho_t, \phi_t, t] \right) dt$$

$$\overset{(ii)}{=} \inf_{\rho_t \in \Gamma(\mu_0, \mu_1)} \sup_{\phi_t} \int \phi_1 \mu_1 dx_1 - \int \phi_0 \mu_0 dx_0 - \int_0^1 \left( \int \frac{\partial \phi_t}{\partial t} \rho_t \, dx_t + \mathcal{H}[\rho_t, \phi_t, t] \right) dt$$

which is the desired result. In (ii), we use the fact that $\rho_0 = \mu_0, \rho_1 = \mu_1$ for $\rho_t \in \Gamma(\mu_0, \mu_1)$. Finally, note that $\phi_t \in \mathcal{T}_{\rho_t}^* \mathcal{P}$ simply identifies $\phi_t$ as a cotangent vector and does not impose meaningful constraints on the form of $\phi_t \in \mathcal{C}^\infty(\mathcal{X})$, so we drop this from the optimization in step (i). □

### A.1    MULTIPLE MARGINAL CONSTRAINTS

Consider multiple marginal constraints in the Lagrangian action minimization problem for $\mathcal{L}[\rho_t, \dot\rho_t, t]$ strictly convex in $\dot\rho_t$,

$$\mathcal{S}_{\mathcal{L}}(\{\mu_{t_i}\}_{i=0}^{M-1}) = \inf_{\rho_t} \int_0^1 \mathcal{L}[\rho_t, \dot\rho_t, t] dt \quad \text{s.t.} \quad \rho_{t_i} = \mu_{t_i} \, (\forall \, 0 \le i \le M - 1) \quad (29)$$

$$= \inf_{\rho_t \in \Gamma(\{\mu_{t_i}\})} \int_0^1 \mathcal{L}[\rho_t, \dot\rho_t, t] dt$$

As in the proof of Thm. 1, the dual becomes

$$\mathcal{S}_{\mathcal{L}}(\{\mu_{t_i}\}_{i=0}^{M-1}) = \inf_{\rho_t \in \Gamma(\{\mu_{t_i}\})} \sup_{\phi_t \in \mathcal{T}_{\rho_t}^* \mathcal{P}} \int_0^1 \left( \int \phi_t \dot{\rho}_t \, dx_t - \mathcal{H}[\rho_t, \phi_t, t] \right) dt$$

$$= \inf_{\rho_t \in \Gamma(\{\mu_{t_i}\})} \sup_{\phi_t} \int \phi_1 \rho_1 dx_1 - \int \phi_0 \rho_0 dx_0 - \int_0^1 \left( \int \frac{\partial \phi_t}{\partial t} \rho_t \, dx_t + \mathcal{H}[\rho_t, \phi_t, t] \right) dt$$

$$= \inf_{\rho_t \in \Gamma(\{\mu_{t_i}\})} \sup_{\phi_t} \int \phi_1 \mu_1 dx_1 - \int \phi_0 \mu_0 dx_0 - \int_0^1 \left( \int \frac{\partial \phi_t}{\partial t} \rho_t \, dx_t + \mathcal{H}[\rho_t, \phi_t, t] \right) dt$$

where the intermediate marginal constraints do not affect the result. Crucially, as discussed in Sec. 3.2.2, our sampling approach satisfies the marginal constraints $\rho_{t_i}(x_{t_i}) = \mu_{t_i}(x_{t_i})$ by design.

**Piecewise Lagrangian Optimization**  Note that the concatenation of dual objectives for $M = 3$, or action-minimization problems between $\{\mu_{0,t_1}\}$ and $\{\mu_{t_1,1}\}$ yields the following dual objective

$$\mathcal{S}_{\mathcal{L}}(\{\mu_{0,t_1}\}) + \mathcal{S}_{\mathcal{L}}(\{\mu_{t_1,1}\}) \qquad (30)$$

$$= \inf_{\rho_t \in \Gamma(\{\mu_0, \mu_{t_1}\})} \sup_{\phi_t} \int \phi_{t_1} \mu_{t_1} dx_{t_1} - \int \phi_0 \mu_0 dx_0 + \int_0^{t_1} \left( \int \frac{\partial \phi_t}{\partial t} \rho_t \, dx_t + \mathcal{H}[\rho_t, \phi_t, t] \right) dt$$

$$+ \inf_{\rho_t \in \Gamma(\{\mu_{t_1}, \mu_1\})} \sup_{\phi_t} \int \phi_1 \mu_1 dx_1 - \int \phi_{t_1} \mu_{t_1} dx_{t_1} + \int_{t_1}^1 \left( \int \frac{\partial \phi_t}{\partial t} \rho_t \, dx_t + \mathcal{H}[\rho_t, \phi_t, t] \right) dt$$

After telescoping cancellation and taking the union of the constraints, we see that our computational approach yields a piece-wise solution to the multi-marginal problem, with $\mathcal{S}_{\mathcal{L}}(\{\mu_{t_i}\}_{i=0}^{M-1}) = \sum_{i=0}^{M-2} \mathcal{S}_{\mathcal{L}}(\{\mu_{t_i, t_{i+1}}\})$.

# B  TRACTABLE OBJECTIVES FOR SPECIAL CASES

In this section, we calculate Hamiltonians and explicit dual objectives for important special cases of Wasserstein Lagrangian Flows, including those in Sec. 4.

We consider several important kinetic energies in App. B.1, including the $W_2$ and $WFR_\lambda$ metrics (App. B.1.1) and the case of OT costs defined by general ground-space Lagrangians (App. B.1.2). In App. B.2, we provide further derivations to obtain a linear dual objective for the Schrödinger Bridge problem. Finally, we highlight the lack of dual linearizability for the case of the Schrödinger Equation App. B.3 Ex. B.3.

## B.1  DUAL KINETIC ENERGY FROM $W_2$, $WFR$, OR GROUND-SPACE LAGRANGIAN COSTS

Thm. 1 makes progress toward a dual objective *without* considering the continuity equation or dynamics in the ground space, by instead invoking the Legendre transform $\mathcal{H}[\rho_t, \phi_t, t]$ of a given Lagrangian $\mathcal{L}[\rho_t, \dot{\rho}_t, t]$ which is strictly convex in $\dot{\rho}_t$. However, to derive $\mathcal{H}[\rho_t, \phi_t, t]$ and optimize objectives of the form Eq. (13), we will need to represent the tangent vector on the space of densities $\dot{\rho}_t$, for example using a vector field $v_t$ and growth term $g_t$ as in Eq. (5).

Given a Lagrangian $\mathcal{L}[\rho_t, \dot{\rho}_t, t]$, we seek to solve the optimization

$$\mathcal{H}[\rho_t, \phi_t, t] = \sup_{\dot{\rho}_t \in \mathcal{T}_{\rho_t} \mathcal{P}} \int \phi_t \dot{\rho}_t \, dx_t - \mathcal{L}[\rho_t, \dot{\rho}_t, t] = \sup_{\dot{\rho}_t \in \mathcal{T}_{\rho_t} \mathcal{P}} \int \phi_t \dot{\rho}_t \, dx_t - \mathcal{K}[\rho_t, \dot{\rho}_t, t] + \mathcal{U}[\rho_t, t]$$

$$(31)$$

Since the potential energy does not depend on $\dot{\rho}_t$, we focus on *kinetic energies* $\mathcal{K}[\rho_t, \dot{\rho}_t, t]$ which are linear in the density (see Def. 3.1). We consider two primary examples, the $WFR_\lambda$ metric $\mathcal{K}[\rho_t, \dot{\rho}_t, t]$ using the continuity equation with growth term dynamics, and kinetic energies defined by expectations of ground-space Lagrangian costs under $\rho_t$ (see App. B.1.2, Villani (2009) Ch. 7, Ex. B.1 below),

$$WFR_\lambda : \quad \mathcal{K}[\rho_t, \dot{\rho}_t, t] = \int \left( \frac{1}{2} \|v_t\|^2 + \frac{\lambda}{2} g_t^2 \right) \rho_t dx_t, \qquad \dot{\rho}_t = -\nabla \cdot (\rho_t v_t) + \lambda \rho_t g_t \quad (32)$$

$$L(\gamma_t, \dot{\gamma}_t, t) : \quad \mathcal{K}[\rho_t, \dot{\rho}_t, t] = \int L(x_t, v_t, t) \rho_t dx_t, \qquad \dot{\rho}_t = -\nabla \cdot (\rho_t v_t) \quad (33)$$

where $(x_t, v_t) = (\gamma_t, \dot{\gamma}_t)$ and we recover the $W_2$ kinetic energy for $L[x_t, v_t, t] = \frac{1}{2}\|v_t\|^2$ or $\lambda = 0$.

We proceed with common derivations, writing $\mathcal{K}[\rho_t, \dot{\rho}_t, t] = \int K(x_t, v_t, g_t, t)\rho_t dx_t$ and simplifying Eq. (31) using the more general dynamics in Eq. (32)

$$\mathcal{H}[\rho_t, \phi_t, t] = \sup_{\dot{\rho}_t \in \mathcal{T}_{\rho_t}\mathcal{P}} \int \phi_t \dot{\rho}_t \ dx_t - \mathcal{K}[\rho_t, \dot{\rho}_t, t] + \mathcal{U}[\rho_t, t] \tag{34}$$

$$= \sup_{(v_t, g_t)} \int \phi_t\big(-\nabla \cdot (\rho_t v_t) + \lambda \rho_t g_t\big) \ dx_t - \mathcal{K}[\rho_t, \dot{\rho}_t, t] + \mathcal{U}[\rho_t, t] \tag{35}$$

Integrating by parts, we have

$$= \sup_{(v_t, g_t)} \int \big(\langle \nabla \phi_t, v_t \rangle \rho_t + \lambda \rho_t \phi_t g_t\big) \ dx_t - \mathcal{K}[\rho_t, \dot{\rho}_t, t] + \mathcal{U}[\rho_t, t]. \tag{36}$$

We now focus on the special cases in Eq. (32) and Eq. (33).

### B.1.1 Wasserstein Fisher-Rao and $W_2$

For $\mathcal{K}[\rho_t, \dot{\rho}_t, t] = \int \big(\frac{1}{2}\|v_t\|^2 + \frac{\lambda}{2}g_t^2\big)\rho_t dx_t$, we proceed from Eq. (36),

$$\mathcal{H}[\rho_t, \phi_t, t] = \sup_{(v_t, g_t)} \int \big(\langle \nabla \phi_t, v_t \rangle \rho_t + \lambda \rho_t \phi_t g_t\big) \ dx_t - \int \Big(\frac{1}{2}\|v_t\|^2 + \frac{\lambda}{2}g_t^2\Big)\rho_t dx_t + \mathcal{U}[\rho_t, t] \tag{37}$$

Eliminating $v_t$ and $g_t$ implies

$$v_t = \nabla \phi_t \qquad g_t = \phi_t \tag{38}$$

where $v_t = \nabla \phi_t$ also holds for the $W_2$ case with $\lambda = 0$. Substituting into Eq. (37), we obtain a Hamiltonian with a dual kinetic energy $\mathcal{K}^*[\rho_t, \dot{\rho}_t, t]$ below that is linear in $\rho_t$ and matches the metric expressed in the cotangent space $\frac{1}{2}\langle \phi_t, \phi_t \rangle_{T_{\rho_t}}^{WFR_\lambda}$,

$$\mathcal{H}[\rho_t, \phi_t, t] = \int \Big(\frac{1}{2}\|\nabla \phi_t\|^2 + \frac{\lambda}{2}\phi_t^2\Big)\rho_t \ dx_t + \mathcal{U}[\rho_t, t] = \frac{1}{2}\langle \phi_t, \phi_t \rangle_{T_{\rho_t}}^{WFR_\lambda} + \mathcal{U}[\rho_t, t]. \tag{39}$$

We make a similar conclusion for the $W_2$ metric with $\lambda = 0$, where the dual kinetic energy is $\mathcal{K}^*[\rho_t, \dot{\rho}_t, t] = \frac{1}{2}\langle \phi_t, \phi_t \rangle_{T_{\rho_t}}^{W_2} = \frac{1}{2}\int \|\nabla \phi_t\|^2 \rho_t \ dx_t$.

### B.1.2 Lifting Ground-Space Lagrangian Costs to Kinetic Energies

We first consider using Lagrangians in the ground space to define costs associated with action-minimizing curves $\gamma^*(x_0, x_1)$. As in Villani (2009) Thm. 7.21, we can consider using this cost to define an optimal transport costs between densities. We show that this corresponds to a special case of our Wasserstein Lagrangian Flows framework with kinetic energy $\mathcal{K}[\rho_t, \dot{\rho}_t, t] = \int L(x_t, v_t, t)\rho_t dx_t$ as in Eq. (33). However, as discussed in Sec. 3, defining our Lagrangians $\mathcal{L}[\rho_t, \dot{\rho}_t, t]$ *directly* on the space of densities allows for more generality using kinetic energies which include growth terms or potential energies which depend on the density.

**Lagrangian and Hamiltonian Mechanics in the Ground-Space** We begin by reviewing action-minimizing curves in the ground space, which forms the basis the Lagrangian formulation of classical mechanics (Arnol'd, 2013). For curves $\gamma(t) : [0, 1] \to \mathcal{X}$ with velocity $\dot{\gamma}_t \in \mathcal{T}_{\gamma(t)}\mathcal{X}$, we consider evaluating a Lagrangian function $L(\gamma_t, \dot{\gamma}_t, t)$ along the curve to define the *action* as the time integral $\mathcal{A}(\gamma) = \int_0^1 L(\gamma_t, \dot{\gamma}_t, t)dt$. Given two endpoints $x_0, x_1 \in \mathcal{X}$, we consider minimizing the action along all curves with the appropriate endpoints $\gamma \in \Pi(x_0, x_1)$,

$$c(x_0, x_1) = \inf_{\gamma \in \Pi(x_0, x_1)} \mathcal{A}(\gamma) = \inf_{\gamma_t} \int_0^1 L(\gamma_t, \dot{\gamma}_t, t)dt \quad \text{s.t. } \gamma_0 = x_0, \ \gamma_1 = x_1 \tag{40}$$

We refer to the optimizing curves $\gamma^*(x_0, x_1)$ as *Lagrangian flows* in the ground-space, which satisfy the Euler-Lagrange equation $\frac{d}{dt}\frac{\partial}{\partial \dot{\gamma}_t}L(\gamma_t, \dot{\gamma}_t, t) = \frac{d}{d\gamma_t}L(\gamma_t, \dot{\gamma}_t, t)$ as a stationarity condition.

We will assume that $L(\gamma_t, \dot{\gamma}_t, t)$ is strictly convex in the velocity $\dot{\gamma}_t$, in which case we can obtain an equivalent, *Hamiltonian* perspective via convex duality. Considering momentum variables $p_t$, we define the Hamiltonian $H(\gamma_t, p_t, t)$ as the Legendre transform of $L$ with respect to $\dot{\gamma}_t$,

$$H(\gamma_t, p_t, t) = \sup_{\dot{\gamma}_t} \ \langle \dot{\gamma}_t, p_t \rangle - L(\gamma_t, \dot{\gamma}_t, t) \tag{41}$$

The Euler-Lagrange equations can be written as Hamilton's equations in the phase space

$$\dot{\gamma}_t = \frac{\partial}{\partial p_t} H(\gamma_t, p_t, t) \qquad \dot{p}_t = -\frac{\partial}{\partial \gamma_t} H(\gamma_t, p_t, t). \tag{42}$$

We proceed to consider Lagrangian actions in the ground-space as a way to construct optimal transport costs over distributions.

**Example B.1** (Ground-Space Lagrangians as OT Costs). The cost function $c(x_0, x_1)$ is a degree of freedom in specifying an optimal transport distance between probability densities $\mu_0, \mu_1 \in \mathcal{P}(\mathcal{X})$ in Eq. (1). Beyond $c(x_0, x_1) = \|x_0 - x_1\|^2$, one might consider defining the OT problem using a cost $c(x_0, x_1)$ induced by a Lagrangian $L(\gamma_t, \dot{\gamma}_t, t)$ in the ground space $\gamma_t \in \mathcal{X}$, as in Eq. (40) (Villani (2009) Ch. 7). In particular, a coupling $\pi(x_0, x_1)$ should assign mass to endpoints $(x_0, x_1)$ based on the Lagrangian cost of their action-minimizing curves $\gamma^*(x_0, x_1)$ Translating to a dynamical formulation (Villani (2009) Thm. 7.21) and using notation $(\gamma_t, \dot{\gamma}_t) = (x_t, v_t)$, the OT problem is

$$W_L(\mu_0, \mu_1) = \inf_{(x_t, v_t)} \int_0^1 \int L(x_t, v_t, t) \rho_t dx_t dt \quad \text{s.t. } \text{law}(x_t) = \rho_t, \ \text{law}(x_0) = \mu_0, \ \text{law}(x_1) = \mu_1. \tag{43}$$

which we may also view as an optimization over the distribution of marginals $\rho_t$ under which $x_t$ is evaluated (see, e.g. Schachter (2017) Def. 3.4.1)

$$W_L(\mu_0, \mu_1) = \inf_{\rho_t} \inf_{v_t} \int_0^1 \int L(x_t, v_t, t) \rho_t dx_t dt \quad \text{s.t. } \dot{\rho}_t = -\nabla \cdot (\rho_t v_t), \ \rho_0 = \mu_0, \ \rho_1 = \mu_1. \tag{44}$$

We can thus view the OT problem as 'lifting' the Lagrangian cost on the ground space $\mathcal{X}$ to a distance in the space of probability densities $\mathcal{P}_2(\mathcal{X})$ via the kinetic energy $\mathcal{K}[\rho_t, \dot{\rho}_t] = \int L(x_t, v_t, t) \rho_t dx_t$ (see below). Of course, the Benamou-Brenier dynamical formulation of $W_2$-OT in Eq. (2) may be viewed as a special case with $L(\gamma_t, \dot{\gamma}_t, t) = L(x_t, v_t, t) = \frac{1}{2}\|v_t\|$.

**Wasserstein Lagrangian and Hamiltonian Perspective** Recognizing the similarity with the Benamou-Brenier formulation in Ex. 4.1, we consider the Wasserstein Lagrangian optimization with two endpoint marginal constraints,

$$\mathcal{S}_{\mathcal{L}}(\{\mu_{0,1}\}) = \inf_{\rho_t \in \Gamma(\mu_0, \mu_1)} \int_0^1 \mathcal{K}[\rho_t, \dot{\rho}_t, t] - \mathcal{U}[\rho_t, t] dt \tag{45}$$

$$= \inf_{\rho_t} \int_0^1 \left( \int L(x_t, v_t, t) \rho_t dx_t - \mathcal{U}[\rho_t, t] \right) dt \quad \text{s.t.} \quad \rho_0 = \mu_0, \qquad \rho_1 = \mu_1$$

Parameterizing the tangent space using the continuity equation as in Eq. (33) or Eq. (44), we can derive the Wasserstein Hamiltonian from Eq. (36) with $\lambda = 0$ (no growth dynamics). Including a potential energy $\mathcal{U}[\rho_t, t]$, we have

$$\mathcal{H}[\rho_t, \phi_t, t] = \sup_{v_t} \ \int \langle \nabla \phi_t, v_t \rangle \rho_t \ dx_t - \mathcal{K}[\rho_t, \dot{\rho}_t, t] + \mathcal{U}[\rho_t, t]. \tag{46}$$

$$= \sup_{v_t} \ \int \langle \nabla \phi_t, v_t \rangle \rho_t \ dx_t - \int L(x_t, v_t, t) \rho_t dx_t + \mathcal{U}[\rho_t, t] \tag{47}$$

$$= \int \left( \sup_{v_t} \ \langle \nabla \phi_t, v_t \rangle - L(x_t, v_t, t) \right) \rho_t \ dx_t + \mathcal{U}[\rho_t, t] \tag{48}$$

which is simply a Legendre transform between velocity and momentum variables in the ground space (Eq. (41)). We can finally write,

$$\mathcal{H}[\rho_t, \phi_t, t] = \int H(x_t, \nabla \phi_t, t) \rho_t dx_t + \mathcal{U}[\rho_t, t] \tag{49}$$

which implies the dual kinetic energy is simply the expectation of the Hamiltonian $\mathcal{K}^*[\rho_t, \phi_t, t] = \int H(x_t, \nabla \phi_t, t) \rho_t dx_t$ and is clearly linear in the density $\rho_t$.

We leave empirical exploration of various Lagrangian costs for future work, but note that $H(x_t, \nabla \phi_t, t)$ in Eq. (49) must be known or optimized using Eq. (48) to obtain a tractable objective.

## B.2 SCHRÖDINGER BRIDGE

In this section, we derive potential energies and tractable objectives corresponding to the Schrödinger Bridge problem

$$S_{SB} = \inf_{\rho_t, v_t} \int_0^1 \int \frac{1}{2} \|v_t\|^2 \rho_t dx_t dt \quad \text{s.t.} \quad \dot{\rho}_t = -\nabla \cdot (\rho_t v_t) - \frac{\sigma^2}{2} \Delta \rho_t \quad \rho_0 = \mu_0, \ \rho_1 = \mu_1. \quad (50)$$

which we will solve using the following (linear in $\rho_t$) dual objective from Eq. (25)

$$\mathcal{S}_{SB} = \inf_{\rho_t \in \Gamma(\mu_0, \mu_1)} \sup_{\Phi_t} \int \Phi_1 \mu_1 dx_1 - \int \Phi_0 \mu_0 dx_0 - \int_0^1 \int \left( \frac{\partial \Phi_t}{\partial t} + \frac{1}{2} \|\nabla \Phi_t\|^2 + \frac{\sigma^2}{2} \Delta \Phi_t \right) \rho_t dx_t dt.$$

**Lagrangian and Hamiltonian for SB** We consider a potential energy of the form,

$$\mathcal{U}[\rho_t, t] = -\frac{\sigma^4}{8} \int \|\nabla \log \rho_t\|^2 \rho_t dx_t \quad (51)$$

which, alongside the $W_2$ kinetic energy, yields the full Lagrangian

$$\mathcal{L}[\rho_t, \dot{\rho}_t, t] = \frac{1}{2} \langle \dot{\rho}_t, \dot{\rho}_t \rangle_{T_{\rho_t}}^{W_2} + \frac{\sigma^4}{8} \int \|\nabla \log \rho_t\|^2 \rho_t dx_t. \quad (52)$$

As in Eq. (34)-(37), we parameterize the tangent space using the continuity equation $\dot{\rho}_t = -\nabla \cdot (\rho_t v_t)$ and vector field $v_t$ in solving for the Hamiltonian,

$$\mathcal{H}[\rho_t, \phi_t, t] = \sup_{\dot{\rho}_t} \int \phi_t \dot{\rho}_t dx_t - \mathcal{L}[\rho_t, \dot{\rho}_t, t] \quad (53)$$

$$= \sup_{v_t} \int \langle \nabla \phi_t, v_t \rangle \rho_t dx_t - \frac{1}{2} \int \|v_t\|^2 \rho_t dx_t - \frac{\sigma_t^4}{8} \int \|\nabla \log \rho_t\|^2 \rho_t dx_t + \int \left( \frac{\partial}{\partial t} \frac{\sigma_t^2}{2} \right) \log \rho_t \, \rho_t dx_t$$

which implies $v_t = \nabla \phi_t$ as before. Substituting into the above, the Hamiltonian becomes

$$\mathcal{H}[\rho_t, \phi_t, t] = \frac{1}{2} \int \|\nabla \phi_t\|^2 \rho_t dx_t - \frac{\sigma^4}{8} \int \|\nabla \log \rho_t\|^2 \rho_t dx_t. \quad (54)$$

which is of the form $\mathcal{H}[\rho_t, \phi_t, t] = \mathcal{K}^*[\rho_t, \phi_t, t] + \mathcal{U}[\rho_t, t]$ and matches Léger & Li (2021) Eq. 8. As in Thm. 1, the dual for the Wasserstein Lagrangian Flow with the Lagrangian in Eq. (52) involves the Hamiltonian in Eq. (54),

$$\mathcal{S}_{\mathcal{L}} = \inf_{\rho_t \in \Gamma(\mu_0, \mu_1)} \sup_{\phi_t} \int \phi_1 \mu_1 dx_1 - \int \phi_0 \mu_0 dx_0 - \int_0^1 \int \left( \frac{\partial \phi_t}{\partial t} + \frac{1}{2} \|\nabla \phi_t\|^2 - \frac{\sigma^4}{8} \int \|\nabla \log \rho_t\|^2 \right) \rho_t \, dx_t$$

$$(55)$$

However, this objective is nonlinear in $\rho_t$ and requires access to $\nabla \log \rho_t$. To linearize the dual objective, we proceed using a reparameterization in terms of the Fokker-Planck equation, or using the Hopf-Cole transform, in the following proposition.

**Proposition 3.** *The solution to the Wasserstein Lagrangian flow*

$$\mathcal{S}_{\mathcal{L}}(\{\mu_{0,1}\}) = \inf_{\rho_t} \int_0^1 \mathcal{L}[\rho_t, \dot{\rho}_t, t] dt \qquad s.t. \qquad \rho_0 = \mu_0, \qquad \rho_1 = \mu_1 \quad (56)$$

*where* $\mathcal{K}[\rho_t, \dot{\rho}_t, t] = \frac{1}{2} \langle \dot{\rho}_t, \dot{\rho}_t \rangle_{T_{\rho_t}}^{W_2}, \qquad \mathcal{U}[\rho_t, t] = -\frac{\sigma^4}{8} \|\nabla \log \rho_t\|_{T_{\rho_t}^{W_2}}^2$

*matches the solution to the* SB *problem in Eq. (50),* $\mathcal{S} = \mathcal{S}_{SB}(\{\mu_{0,1}\}) = \mathcal{S}_{\mathcal{L}}(\{\mu_{0,1}\}) + c(\{\mu_{0,1}\})$ *up to a constant* $c(\{\mu_{0,1}\})$ *wrt* $\rho_t$.

*Further,* $\mathcal{S}$ *is the solution to the (dual) optimization*

$$\mathcal{S} = \inf_{\rho_t \in \Gamma(\mu_0, \mu_1)} \sup_{\Phi_t} \int \Phi_1 \mu_1 dx_1 - \int \Phi_0 \mu_0 dx_0 - \int_0^1 \int \left( \frac{\partial \Phi_t}{\partial t} + \frac{1}{2} \|\nabla \Phi_t\|^2 + \frac{\sigma^2}{2} \Delta \Phi_t \right) \rho_t dx_t dt. \quad (57)$$

*Thus, we obtain a dual objective for the SB problem, or WLF in Eq. (56), which is linear in* $\rho_t$.

*Proof.* We consider the following reparameterization (Léger & Li, 2021)

$$\phi_t = \Phi_t - \frac{\sigma^2}{2} \log \rho_t, \qquad\qquad \nabla \phi_t = \nabla \Phi_t - \frac{\sigma^2}{2} \nabla \log \rho_t. \qquad (58)$$

Note that $\phi_t$ is the drift for the continuity equation in Eq. (53), $\dot{\rho}_t = -\nabla \cdot (\rho_t \nabla \phi_t)$. Via the above reparameterization, we see that $\nabla \Phi_t$ corresponds to the drift in the Fokker-Planck dynamics $\dot{\rho}_t = -\nabla \cdot (\rho_t \nabla \Phi_t) + \frac{\sigma^2}{2} \nabla \cdot (\rho_t \nabla \log \rho_t) = -\nabla \cdot (\rho_t \nabla \Phi_t) + \frac{\sigma^2}{2} \Delta \rho_t$.

*Wasserstein Lagrangian Dual Objective after Reparameterization:* Starting from the dual objective in Eq. (55), we perform the reparameterization in Eq. (58), $\phi_t = \Phi_t - \frac{\sigma^2}{2} \log \rho_t$,

$$\mathcal{S}_\mathcal{L} = \inf_{\rho_t \in \Gamma(\mu_0, \mu_1)} \sup_{\Phi_t} \int \Phi_1 \mu_1 dx_1 - \frac{\sigma^2}{2} \int \log \rho_1 \, \mu_1 dx_1 - \int \Phi_0 \mu_0 dx_0 + \frac{\sigma^2}{2} \int \log \rho_0 \, \mu_0 dx_0 \qquad (59)$$

$$- \int_0^1 \int \left( \frac{\partial \Phi_t}{\partial t} + \frac{\partial}{\partial t} \left( \frac{\sigma^2}{2} \log \rho_t \right) + \frac{1}{2} \left\langle \nabla \Phi_t - \frac{\sigma^2}{2} \nabla \log \rho_t, \nabla \Phi_t - \frac{\sigma^2}{2} \nabla \log \rho_t \right\rangle - \frac{\sigma^4}{8} \|\nabla \log \rho_t\|^2 \right) \rho_t \, dx_t dt$$

Noting that the $\int \frac{\sigma^2}{2} (\frac{\partial}{\partial t} \log \rho_t) \rho_t dx_t$ cancels since $\frac{\partial}{\partial t} \int \rho_t dx_t = 0$, we simplify to obtain

$$\mathcal{S}_\mathcal{L} = \inf_{\rho_t \in \Gamma(\mu_0, \mu_1)} \sup_{\Phi_t} \int \Phi_1 \mu_1 dx_1 - \frac{\sigma^2}{2} \int \log \rho_1 \, \mu_1 dx_1 - \int \Phi_0 \mu_0 dx_0 + \frac{\sigma^2}{2} \int \log \rho_0 \mu_0 dx_0$$

$$- \int_0^1 \int \left( \frac{\partial \Phi_t}{\partial t} + \frac{1}{2} \|\nabla \Phi_t\|^2 - \frac{\sigma^2}{2} \langle \nabla \Phi_t, \nabla \log \rho_t \rangle \right) \rho_t dx_t dt$$

where the Hamiltonian now matches Eq. 7 in Léger & Li (2021). Taking $\nabla \log \rho_t = \frac{1}{\rho_t} \nabla \rho_t$ and integrating by parts, the final term becomes

$$\mathcal{S}_\mathcal{L} = \inf_{\rho_t \in \Gamma(\mu_0, \mu_1)} \sup_{\Phi_t} \int \Phi_1 \mu_1 dx_1 - \frac{\sigma^2}{2} \int \log \rho_1 \, \mu_1 dx_1 - \int \Phi_0 \mu_0 dx_0 + \frac{\sigma^2}{2} \int \log \rho_0 \mu_0 dx_0$$

$$- \int_0^1 \int \left( \frac{\partial \Phi_t}{\partial t} + \frac{1}{2} \|\nabla \Phi_t\|^2 + \frac{\sigma^2}{2} \Delta \Phi_t \right) \rho_t dx_t dt$$

Finally, we consider adding terms $c(\{\mu_{0,1}\}) = \frac{\sigma^2}{2} \int \log \mu_1 \, \mu_1 dx_1 - \frac{\sigma^2}{2} \int \log \mu_0 \, \mu_0 dx_0$ which are constant with respect to $\rho_{0,1}$,

$$\mathcal{S}_\mathcal{L}(\{\mu_{0,1}\}) + c(\{\mu_{0,1}\}) = \inf_{\rho_t \in \Gamma(\mu_0, \mu_1)} \sup_{\Phi_t} \int \Phi_1 \mu_1 dx_1 + \frac{\sigma_1^2}{2} \int (\log \mu_1 \, - \log \rho_1) \mu_1 dx_1 \qquad (60)$$

$$- \int \Phi_0 \mu_0 dx_0 - \frac{\sigma_0^2}{2} \int (\log \mu_0 \, - \log \rho_0) \mu_0 dx_0$$

$$- \int_0^1 \int \left( \frac{\partial \Phi_t}{\partial t} + \frac{1}{2} \|\nabla \Phi_t\|^2 + \frac{\sigma_t^2}{2} \Delta \Phi_t \right) \rho_t dx_t dt$$

Finally, the endpoint terms vanish for $\rho_t \in \Gamma(\mu_0, \mu_1)$ satisfying the endpoint constraints,

$$\mathcal{S}_\mathcal{L}(\{\mu_{0,1}\}) + c(\{\mu_{0,1}\}) \qquad\qquad\qquad\qquad\qquad\qquad\qquad\qquad\qquad\qquad\qquad (61)$$

$$= \inf_{\rho_t \in \Gamma(\mu_0, \mu_1)} \sup_{\Phi_t} \int \Phi_1 \mu_1 dx_1 - \int \Phi_0 \mu_0 dx_0 - \int_0^1 \int \left( \frac{\partial \Phi_t}{\partial t} + \frac{1}{2} \|\nabla \Phi_t\|^2 + \frac{\sigma^2}{2} \Delta \Phi_t \right) \rho_t dx_t dt$$

which matches the dual in Eq. (57). We now show that this is also the dual for the SB problem.

*Schrödinger Bridge Dual Objective:* Consider the optimization in Eq. (50) (here, $t$ may be time-dependent)

$$\mathcal{S}_{SB}(\{\mu_{0,1}\}) = \inf_{\rho_t, v_t} \int_0^1 \frac{1}{2} \|v_t\|^2 \rho_t dx_t \quad \text{s.t.} \quad \dot{\rho}_t = -\nabla \cdot (\rho_t v_t) + \frac{\sigma_t^2}{2} \nabla \cdot (\rho_t \nabla \log \rho_t), \, \rho_0 = \mu_0, \, \rho_1 = \mu_1$$

$$(62)$$

We treat the optimization over $\rho_t$ as an optimization over a vector space of functions, which is later constrained be normalized via the $\rho_0 = \mu_0, \rho_1 = \mu_1$ constraints and continuity equation (which

preserves normalization). It is also constrained to be nonnegative, but we omit explicit constraints for simplicity of notation. The optimization over $v_t$ is also over a vector space of functions. See App. C for additional discussion.

Given these considerations, we may now introduce Lagrange multipliers $\lambda_0, \lambda_1$ to enforce the end-point constraints and $\Phi_t$ to enforce the dynamics constraint,

$$\mathcal{S}_{SB}(\{\mu_{0,1}\}) = \inf_{\rho_t,v_t} \sup_{\Phi_t,\lambda_{0,1}} \int_0^1 \frac{1}{2}\|v_t\|^2 \rho_t dx_t + \int \Phi_t\Big(\dot{\rho}_t + \nabla\cdot(\rho_t v_t) - \frac{\sigma_t^2}{2}\nabla\cdot(\rho_t\,\nabla\log\rho_t)\Big)dx_t \quad (63)$$

$$+ \int \lambda_1(\rho_1 - \mu_1)dx_1 + \int \lambda_0(\rho_0 - \mu_0)dx_0$$

$$= \inf_{\rho_t,v_t} \sup_{\Phi_t,\lambda_{0,1}} \int_0^1 \frac{1}{2}\|v_t\|^2 \rho_t dx_t + \int \Phi_1\rho_1 dx_1 - \int \Phi_0\rho_0 dx_0 - \int_0^1 \int \frac{\partial \Phi_t}{\partial t}\rho_t dx_t dt \quad (64)$$

$$- \int_0^1 \int \Big\langle \nabla\Phi_t, v_t - \frac{\sigma_t^2}{2}\nabla\log\rho_t \Big\rangle \rho_t dx_t dt + \int \lambda_1(\rho_1 - \mu_1)dx_1 + \int \lambda_0(\rho_0 - \mu_0)dx_0$$

Note that we can freely we can swap the order of the optimizations since the SB optimization in Eq. (62) is convex in $\rho_t, v_t$, while the dual optimization is linear in $\Phi_t, \lambda$.

Swapping the order of the optimizations and eliminating $\rho_0$ and $\rho_1$ implies $\lambda_1 = \Phi_1$ and $\lambda_0 = \Phi_0$, while eliminating $v_t$ implies $v_t = \nabla\Phi_t$. Finally, we obtain

$$\mathcal{S}_{SB}(\{\mu_{0,1}\}) = \sup_{\Phi_t} \inf_{\rho_t} \int \Phi_1\mu_1 dx_1 - \int \Phi_0\mu_0 dx_0 - \int_0^1 \Big(\frac{\partial \Phi_t}{\partial t} + \frac{1}{2}\|\Phi_t\|^2 - \frac{\sigma_t^2}{2}\langle\nabla\Phi_t, \nabla\log\rho_t\rangle\Big)\rho_t dx_t$$

$$= \inf_{\rho_t} \sup_{\Phi_t} \int \Phi_1\mu_1 dx_1 - \int \Phi_0\mu_0 dx_0 - \int_0^1 \Big(\frac{\partial \Phi_t}{\partial t} + \frac{1}{2}\|\Phi_t\|^2 + \frac{\sigma_t^2}{2}\Delta\Phi_t\Big)\rho_t dx_t \quad (65)$$

where we swap the order of optimization again in the second line. This matches the dual in Eq. (60) for $\mathcal{S}_{\mathcal{L}}(\{\mu_{0,1}\}) + c(\{\mu_{0,1}\})$ if $\frac{\sigma^2}{2}$ is independent of time, albeit without the endpoint constraints. However, we have shown above that the optimal $\lambda_0^* = \Phi_0^*$, $\lambda_1^* = \Phi_1^*$ will indeed enforce the endpoint constraints. This is the desired result in Proposition 3. $\qquad\square$

**Example B.2** (**Schrödinger Bridge with Time-Dependent Diffusion Coefficient**). To incorporate a time-dependent diffusion coefficient for the classical SB problem, we modify the potential energy with an additional term

$$\mathcal{U}[\rho_t, t] = -\frac{\sigma_t^4}{8}\int \|\nabla\log\rho_t\|^2 \rho_t dx_t + \int \Big(\frac{\partial}{\partial t}\frac{\sigma_t^2}{2}\Big)\log\rho_t\,\rho_t dx_t \quad (66)$$

This potential energy term is chosen carefully to cancel with the term appearing after reparameterization using $\phi_t = \Phi_t - \frac{\sigma_t^2}{2}\log\rho_t$ in Eq. (59). In this case,

$$\int \frac{\partial \phi_t}{\partial t}\rho_t dx_t = \int \Big(\frac{\partial \Phi_t}{\partial t} - \frac{\partial}{\partial t}\Big(\frac{\sigma_t^2}{2}\log\rho_t\Big)\Big)\rho_t\,dx_t \quad (67)$$

$$= \int \Big(\frac{\partial \Phi_t}{\partial t} - \Big(\frac{\partial}{\partial t}\frac{\sigma_t^2}{2}\Big)\log\rho_t + \frac{\sigma_t^2}{2}\Big(\frac{\partial}{\partial t}\log\rho_t\Big)\Big)\rho_t dx_t \quad (68)$$

$$= \int \Big(\frac{\partial \Phi_t}{\partial t} - \Big(\frac{\partial}{\partial t}\frac{\sigma_t^2}{2}\Big)\log\rho_t\Big)\rho_t dx_t \quad (69)$$

where the score term cancels as before. The additional potential energy term is chosen to cancel the remaining term. All other derivations proceed as above, which yields an identical dual objective

$$\mathcal{S}_{SB} = \inf_{\rho_t\in\Gamma(\mu_0,\mu_1)} \sup_{\Phi_t} \int \Phi_1 d\mu_1 - \int \Phi_0 d\mu_0 - \int_0^1 \int \Big(\frac{\partial \Phi_t}{\partial t} + \frac{1}{2}\|\nabla\Phi_t\|^2 + \frac{\sigma_t^2}{2}\Delta\Phi_t\Big)\rho_t dx_t dt$$

### B.3 SCHRÖDINGER EQUATION

**Example B.3** (Schrödinger Equation). Intriguingly, we obtain the Schrödinger Equation via a simple change of sign in the potential energy $\mathcal{U}[\rho_t, t] = \frac{\sigma_t^4}{8}\int \|\nabla\log\rho_t\|^2 \rho_t dx_t$ compared to Eq. (51)

or, in other words, an imaginary weighting $i\sigma_t$ of the gradient norm of the Shannon entropy,

$$\mathcal{L}[\rho_t, \dot{\rho}_t, t] = \frac{1}{2}\langle \dot{\rho}_t, \dot{\rho}_t \rangle_{T_{\rho_t}}^{W_2} - \int \left[ \frac{1}{8}\|\nabla \log \rho_t\|^2 + V_t(x_t) \right] \rho_t \, dx_t \tag{70}$$

This Lagrangian corresponds to a Hamiltonian $\mathcal{H}[\rho_t, \phi_t, t] = \frac{1}{2}\langle \phi_t, \phi_t \rangle_{T_{\rho_t}^*}^{W_2} + \int \left[ \frac{1}{8}\|\nabla \log \rho_t\|^2 + V_t(x_t) \right] \rho_t \, dx_t$, which leads to the dual objective

$$\begin{aligned}
\mathcal{S}_{SE} = \sup_{\phi_t} \inf_{\rho_t} \int \phi_1 d\mu_1 - \int \phi_0 d\mu_0 \\
- \int_0^1 \int \left( \frac{\partial \phi_t}{\partial t} + \frac{1}{2}\|\nabla \phi_t\|^2 + \frac{1}{8}\|\nabla \log \rho_t\|^2 + V_t(x_t) \right) \rho_t dx_t dt.
\end{aligned} \tag{71}$$

Unlike the Schrödinger Bridge problem, the Hopf-Cole transform does not linearize the dual objective in density. Thus, we cannot approximate the dual using only the Monte Carlo estimate.

The first-order optimality conditions for Eq. (71) are

$$\dot{\rho}_t = -\nabla \cdot (\rho_t \nabla \phi_t), \quad \frac{\partial \phi_t}{\partial t} + \frac{1}{2}\|\nabla \phi_t\|^2 = \frac{1}{8}\|\nabla \log \rho_t\|^2 + \frac{1}{4}\Delta \log \rho_t - V_t(x_t) \tag{72}$$

Note, that Eq. (72) is the Madelung transform of the Schrödinger equation, i.e. for the equation

$$\frac{\partial}{\partial t}\psi_t(x) = -i\hat{H}\psi_t(x), \quad \text{where} \quad \hat{H} = -\frac{1}{2}\Delta + V_t(x), \tag{73}$$

the wave function $\psi_t(x)$ can be written in terms $\psi_t(x) = \sqrt{\rho_t(x)} \exp(i\phi_t(x))$. Then the real and imaginary part of the Schrödinger equation yield Eq. (72).

## C  LAGRANGE MULTIPLIER APPROACH

Our Thm. 1 is framed completely in the abstract space of densities and the Legendre transform between functionals of $\dot{\rho}_t \in \mathcal{T}_{\rho_t}\mathcal{P}$ and $\phi_{\dot{\rho}_t} \in \mathcal{T}_{\rho_t}^*\mathcal{P}$. We contrast this approach with optimizations such as the Benamou-Brenier formulation in Eq. (2), which are formulated in terms of the state space dynamics such as the continuity equation $\dot{\rho}_t = -\nabla \cdot (\rho_t v_t)$. In this appendix, we claim that the latter approaches require a potential energy $\mathcal{U}[\rho_t, t]$ which is concave or linear in $\rho_t$. We restrict attention to continuity equation dynamics in this section, although similar reasoning holds with growth terms.

In particular, consider optimizing $\rho_t, v_t$ over a topological vector space of functions. The notable difference here is that $\rho_t : \mathcal{X} \to \mathbb{R}$ is a function, which we later constrain to be a normalized probability density using $\rho_0 = \mu_0, \rho_1 = \mu_1$, the continuity equation $\dot{\rho}_t = -\nabla \cdot (\rho_t v_t)$ (which preserves normalization), and nonnegativity constraints. Omitting the latter for simplicity of notation, we consider the $W_2$ kinetic energy with an arbitrary potential energy,

$$\mathcal{S} = \inf_{\rho_t, v_t} \int_0^1 \int L(x_t, v_t, t)\rho_t dx_t dt - \int_0^1 \mathcal{U}[\rho_t, t]dt \quad \text{s.t.} \quad \dot{\rho}_t = -\nabla \cdot (\rho_t v_t) \quad \rho_0 = \mu_0, \ \rho_1 = \mu_1 \tag{74}$$

Since we are now optimizing $\rho_t$ over a vector space, we introduce Lagrange multipliers $\lambda_{0,1}$ to enforce the endpoint constraints and $\phi_t$ to enforce the continuity equation. Integrating by parts in $t$ and $x$, we have

$$\begin{aligned}
\mathcal{S} = \inf_{\rho_t, v_t} \sup_{\lambda_{0,1}, \phi_t} \int_0^1 \int L(x_t, v_t, t)\rho_t dx_t dt - \int_0^1 \mathcal{U}[\rho_t, t]dt + \int_0^1 \int \phi_t \dot{\rho}_t dx_t + \int_0^1 \int \phi_t \nabla \cdot (\rho_t v_t) dx_t dt \\
+ \int \lambda_0(\rho_0 - \mu_0)dx_0 + \int \lambda_1(\rho_1 - \mu_1)dx_1
\end{aligned} \tag{75}$$

$$\begin{aligned}
= \inf_{\rho_t, v_t} \sup_{\lambda_{0,1}, \phi_t} \int_0^1 \int L(x_t, v_t, t)\rho_t dx_t dt - \int_0^1 \mathcal{U}[\rho_t, t]dt - \int_0^1 \int \frac{\partial \phi_t}{\partial t}\rho_t dx_t - \int_0^1 \int \langle \nabla \phi_t, v_t \rangle \rho_t \, dx_t dt \\
+ \int \lambda_1 \rho_1 dx_1 - \int \lambda_0 \rho_0 dx_0 + \int \lambda_0 \rho_0 dx_0 - \int \lambda_0 \mu_0 \, dx_0 + \int \lambda_1 \rho_1 \, dx_1 - \int \lambda_1 \mu_1 \, dx_1
\end{aligned} \tag{76}$$

To make further progress by swapping the order of the optimizations, we require that Eq. (76) is convex in $\rho_t, v_t$ and concave in $\lambda_{0,1}, \phi_t$. However, to facilitate this, we require that $\mathcal{U}[\rho_t, t]$ is concave in $\rho_t$, which is an additional constraint which was not necessary in the proof of Thm. 1.

By swapping the order of optimization to eliminate $\rho_0, \rho_1$ and $v_t$, we obtain the optimality conditions

$$\lambda_0 = \phi_0, \ \lambda_1 = \phi_1 \qquad v_t = \nabla_p H(x_t, \nabla\phi_t, t) \tag{77}$$

where the gradient is with respect to the second argument. Swapping the order of optimizations again, the dual becomes

$$\mathcal{S} = \inf_{\rho_t} \sup_{\phi_t} \int \phi_1 \mu_1 \, dx_1 - \int \phi_0 \mu_0 \, dx_0 - \int_0^1 \left( \int \left( \frac{\partial \phi_t}{\partial t} + H(x_t, \nabla\phi_t, t) \right) \rho_t dx_t + \mathcal{U}[\rho_t, t] \right) dt.$$

which is analogous to Eq. (13) in Thm. 1 for the $W_2$ kinetic energy. While the dual above does not explicitly enforce the endpoint marginals on $\rho_t$, the conditions $\lambda_0^* = \phi_0^*$, $\lambda_1^* = \phi_1^*$ serve to enforce the constraint at optimality.

## D  EXPRESSIVITY OF PARAMETERIZATION

**Proposition 2.** *For any absolutely-continuous distributional path $\rho_t : [0,1] \mapsto \mathcal{P}_2(\mathcal{X})$ on the $W_2$ manifold, there exists a function $\mathrm{NNET}^*(t, x_0, x_1, \mathbb{1}[t < 0.5]; \eta)$ such that Eq. (16) samples from $\rho_t$.*

*Proof.* For every absolutely-continuous distributional path $\rho_t$, we have a unique gradient flow $\nabla s_t^*(x_t)$ satisfying the continuity equation (Ambrosio et al. (2008) Thm. 8.3.1),

$$\dot{\rho}_t = -\nabla \cdot (\rho_t \nabla s_t^*(x_t)). \tag{78}$$

Consider the function

$$\varphi_t(x_0, x_1) = \begin{cases} x_0 + \int_0^t \nabla s_\tau^*(x_\tau) d\tau, & t \leq 1/2, \\ x_1 + \int_1^t \nabla s_\tau^*(x_\tau) d\tau, & t > 1/2, \end{cases} \tag{79}$$

which integrates the ODE $dx/dt = \nabla s_t^*(x_t)$ forward starting from $x_0$ for $t \leq 1/2$, and integrates the same ODE backwards starting from $x_1$ otherwise.

Clearly, for $t \leq 1/2$ the designed function serves as a push-forward map for the samples $x_0 \sim \rho_0$, and produces samples from $\rho_t$ by Eq. (78). The same applies for $t > 1/2$. Thus, $\varphi_t$ samples from the correct marginals, i.e.

$$\int \delta(x_t - \varphi_t(x_0, x_1)) \rho_0(x_0) \rho_1(x_1) dx_0 dx_1 = \rho_t(x_t), \ \ \forall t \in [0,1]. \tag{80}$$

We now show that $\varphi_t(x_0, x_1)$ can be expressed using the parameterization in Eq. (16), which constructs $x_t$ as

$$x_t = (1-t)x_0 + tx_1 + t(1-t)\mathrm{NNET}^*(t, x_0, x_1, \mathbb{1}[t < 0.5]; \eta), \ \ x_0 \sim \mu_0, \ \ x_1 \sim \mu_1. \tag{81}$$

Then taking the function $\mathrm{NNET}^*(t, x_0, x_1, \mathbb{1}[t < 0.5]; \eta)$ as follows

$$\mathrm{NNET}^*(t, x_0, x_1, \mathbb{1}[t < 0.5]; \eta) = \begin{cases} \frac{1}{1-t}\left(x_0 - x_1 + \frac{1}{t}\int_0^t \nabla s_\tau^*(x_\tau) d\tau\right), & t \leq 1/2, \\ \frac{1}{t}\left(x_1 - x_0 + \frac{1}{1-t}\int_1^t \nabla s_\tau^*(x_\tau) d\tau\right), & t > 1/2, \end{cases} \tag{82}$$

we have

$$(1-t)x_0 + tx_1 + t(1-t)\mathrm{NNET}^*(t, x_0, x_1; \mathbb{1}[t < 0.5]; \eta) = \varphi_t(x_0, x_1), \tag{83}$$

which samples from the correct marginals by construction. $\qquad\square$

# E    DETAILS OF EXPERIMENTS

## E.1    SINGLE-CELL EXPERIMENTS

We consider low dimensional (Table 2) and high dimensional (Table 1) single-cell experiments following the experimental setups in Tong et al. (2023b;a). The Embryoid body (**EB**) dataset Moon et al. (2019) and the CITE-seq (**Cite**) and Multiome (**Multi**) datasets (Burkhardt et al., 2022) are repurposed and preprocessed by Tong et al. (2023b;a) for the task of trajectory inference.

The **EB** dataset is a scRNA-seq dataset of human embryonic stem cells used to observe differentiation of cell lineages (Moon et al., 2019). It contains approximately 16,000 cells (examples) after filtering, of which the first 100 principle components over the feature space (gene space) are used. For the low dimensional (5-dim) experiments, we consider only the first 5 principle components. The **EB** dataset comprises a collection of 5 timepoints sampled over a period of 30 days.

The **Cite** and **Multi** datasets are taken from the Multimodal Single-cell Integration challenge at NeurIPS 2022 (Burkhardt et al., 2022). Both datasets contain single-cell measurements from CD4+ hematopoietic stem and progenitor cells (HSPCs) for 1000 highly variables genes and over 4 timepoints collected on days 2, 3, 4, and 7. We use the **Cite** and **Multi** datasets for both low dimensional (5-dim) and high dimensional (50-dim, 100-dim) experiments. We use 100 computed principle components for the 100-dim experiments, then select the first 50 and first 5 principle components for the 50-dim and 5-dim experiments, respectively. Further details regarding the raw dataset can be found at the competition website. [3]

For all experiments, we train $k$ independent models over $k$ partitions of the single-cell datasets. The training data partition is determined by a left out intermediary timepoint. We then average test performance over the $k$ independent model predictions computed on the respective left-out marginals. For experiments using the **EB** dataset, we train 3 independent models using marginals from timepoint partitions $[1,3,4,5], [1,2,4,5], [1,2,3,5]$ and evaluate each model using the respective left-out marginals at timepoints $[2], [3], [4]$. Likewise, for experiments using **Cite** and **Multi** datasets, we train 2 independent models using marginals from timepoint partitions $[2,4,7], [2,3,7]$ and evaluate each model using the respective left-out marginals at timepoints $[3], [4]$.

For both $\phi_t(x, \theta)$ and $\rho_t(x, \eta)$, we consider Multi-Layer Perceptron (MLP) architectures and a common optimizer (Loshchilov & Hutter, 2017). For detailed description of the architectures and hyperparameters we refer the reader to the code supplemented.

## E.2    SINGLE-STEP IMAGE GENERATION VIA OPTIMAL TRANSPORT

Learning the vector field that corresponds to the optimal transport map between some prior distribution (e.g. Gaussian) and the target data allows to generate data samples evaluating the vector field only once. Indeed, the optimality condition (Hamilton-Jacobi equation) for the dynamical optimal transport yields

$$\ddot{X}_t = \nabla \left[ \frac{\partial \phi_t(x_t)}{\partial t} + \frac{1}{2} \|\nabla \phi_t(x_t)\|^2 \right] = 0 \,, \tag{84}$$

hence, the acceleration along every trajectory is zero. This implies that the learned vector field can be trivially integrated, i.e.

$$X_1 = X_0 + \nabla \phi_0(X_0) \,. \tag{85}$$

Thus, $X_1$ is generated with a single evaluation of $\nabla \phi_0(\cdot)$.

For the image generation experiments, we follow common practices of training the diffusion models (Song et al., 2020), i.e. the vector field model $\phi_t(x, \theta)$ uses the U-net architecture (Ronneberger et al., 2015) with the time embedding and hyperparameters from (Song et al., 2020). For the distribution path model $\rho_t(x, \eta)$, we found that the U-net architectures works best as well. For detailed description of the architectures and hyperparameters we refer the reader to the code supplemented.

---

[3]https://www.kaggle.com/competitions/open-problems-multimodal/data

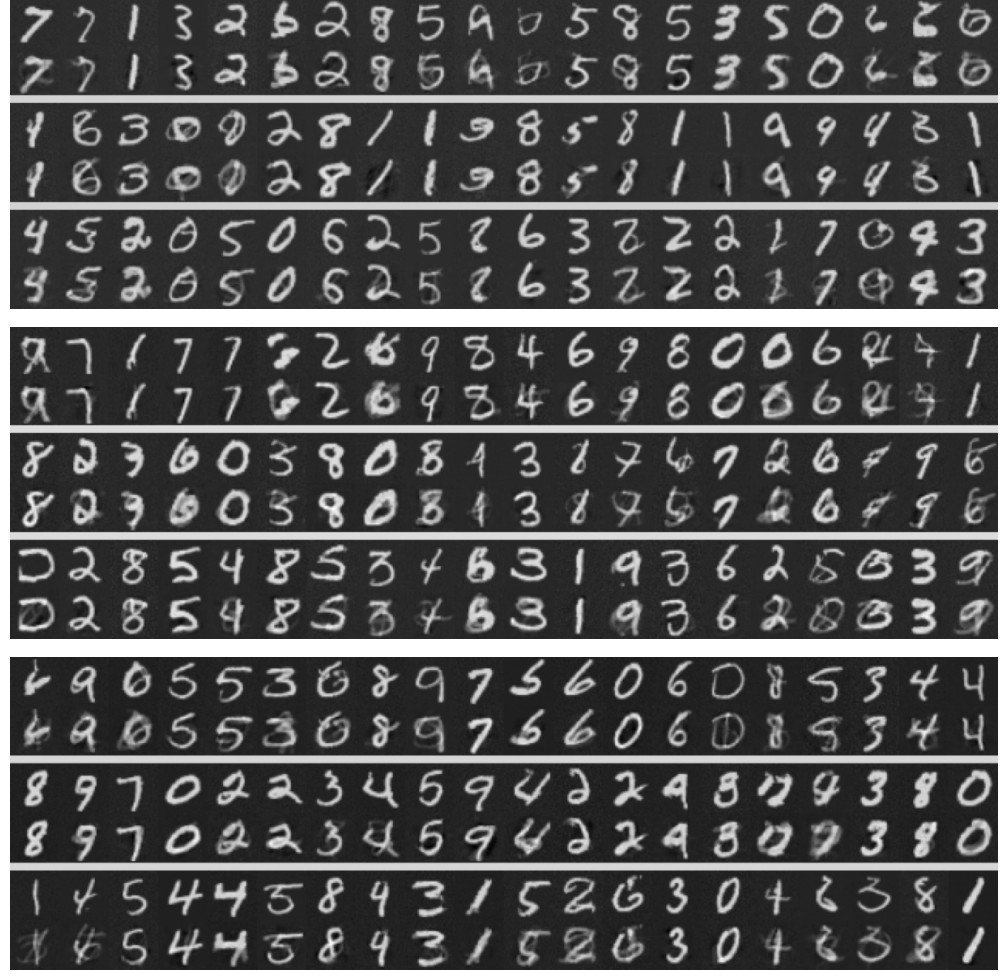

Figure 3: MNIST 32x32 image generation. Every top row is the integration of the corresponding ODE via Dormand-Prince's 5/4 method, which makes 108 function evaluations. Every bottom row corresponds to single function evaluation approximation.

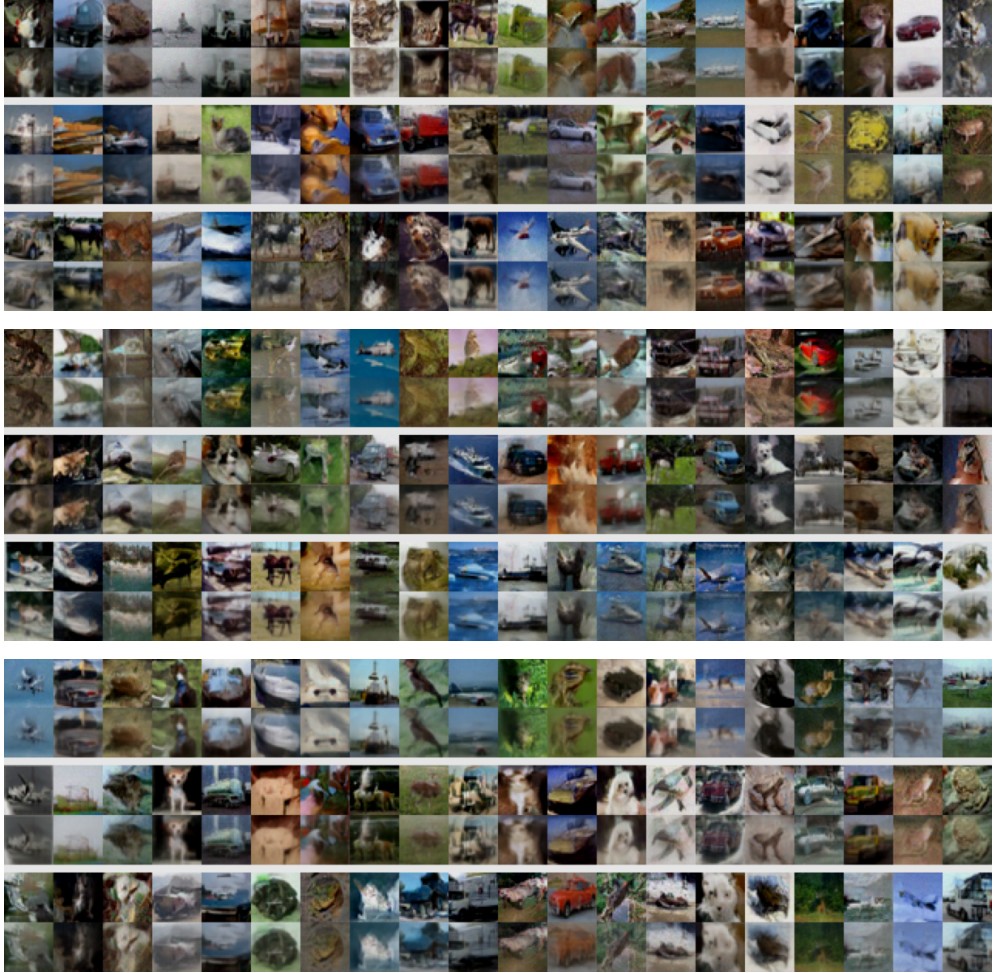

Figure 4: CIFAR-10 image generation. Every top row is the integration of the corresponding ODE via Dormand-Prince's $5/4$ method, which makes $78$ function evaluations. Every bottom row corresponds to single function evaluation approximation.

