# OpenReview forum: "A Computational Framework for Solving Wasserstein Lagrangian Flows"
_ICLR.cc/2024/Conference — Submitted to ICLR 2024_

### Official Review · Reviewer_feJt · 2023-10-20

**Soundness:** 3 good
**Presentation:** 3 good
**Contribution:** 2 fair
**Rating:** 6
**Confidence:** 1

**Summary:**

The authors study the task of optimal transport framing it as an optimization of a lagrangian. They find particular instances where the optimization of this generally hard problem can be performed more efficiently from a dual characterization of the optimization. Their framework works particularly well when the Hamiltonian is linear in the density variable and they give special cases where this occurs. The experiments focus on RNA sequencing tasks for which their method seem to perform well.

**Strengths:**

**Disclaimer:** I am not an expert in optimal transport and have no background in the biological task which appears to be the motivation for this particular work. The AC asked me still to provide a high level review and I will try my best to do so below.  I will refrain from making strong statements about the paper due to my lack of expertise.

The strength of this paper seems to lie in its efficiency in addressing a potentially challenging optimal transport problem. Their reformulation of the optimal transport into an efficient scheme is noteworthy to me. Their algorithm is clearly laid out in section 3 and I could follow along relatively well, albeit many choices such as the neural network architecture are not specified (perhaps to give one flexibility in this choice).

**Weaknesses:**

As stated above, I am not an expert in this area, and reading this paper, I was impressed by the rigor and extensive detail of their approach. Nevertheless, in the end, I could not come to a concrete conclusion regarding its novelty and relevance for someone who is not specifically in their field. Let me explain a bit further.

First, they propose a method which they say is effective and efficient in a wide variety of domains including transport with constraints and unbalanced transport. On this point, my opinion seems to agree with the authors overall, but the specifics contradict the generality at times. For example, the constraints that the authors are able to enforce are constraints on the marginal values themselves. However, what if the constraints come in a different form, e.g. a symmetry in the problem or distributions? This happens all the time in physics and it's not obvious to me that the approach covers a constraint like that.

Second, the authors motivate this problem with a RNA sequencing task saying that we can only observe population statistics and we want to infer individual statistics from this. I can understand this would be useful. But looking at the experiments, none of the experiments test this specific point. All the experiments are predicting population statistics. This may be an issue in how one collects the data rather than the learning of that data, but nonetheless it leaves me unconvinced that this approach actually achieves the high level goal.

Typos/grammar and notational comments:
- pg. 2: "which forms the basis the Lagrangian formulation”
- section 2.1 writes the Lagrangian as inputting path variables, but later on, the inputs to the Lagrangian are momentum variables. This should be consistent.
- pg. 3 grammar: "allows us to consider of potential energy functionals which depend"
- Hyperlinks for OT, SB, etc. seem to be broken. What are these supposed to point to?

Questions:
- Below eq 27, the authors say that they perform optimization with respect to $\eta$ using a reparameterization trick. What is this trick? I couldn’t find a reference to it.
- In the experiments, which time points are the left-out marginals? Are the left-out marginals in the future and need to be extrapolated? Or is it interpolated as in-between points? The former task seems to be a lot harder in my opinion.


Minor comments:
- I wish the authors expanded a bit more about the potential extension to the quantum Schrödinger equation, since this is a field I am particularly interested in. The Madelung transformation in the appendix appears to only be for one particle (unless I am missing something) which is not really of practical relevance. The authors do not state this exactly and I wish they were more precise in how they view this extension in their appendix section. Extending classical densities of particles to work in the quantum setting for multiple particles is a tricky task as it needs to account for factors like entanglement; e.g. see density functional theory which takes on this complication in rather extensive detail.
- The particular task in the experiment is tough to follow unless one goes into the appendix and digs into the details of the Kaggle competition itself. I wish the authors would describe more what exactly is going on. E.g. simple things like what the raw data is in the form of. The resulting task is the PCA of this raw data.
- As stated in my disclaimer above, I am by no means an expert in this topic, but I do not think this paper unifies the field. In fact, there were many situations where the authors stated to their credit that their method may not apply very well (e.g. Schrodinger equation). I am sure people would have to get a different perspective for these problems. So I would caution the authors in their language surrounding this point.

On a final point, given my lack of background, I did not check equations line by line and proofs were not checked.

Overall, I have chosen a score of marginal accept for the reasons stated in my admittedly limited review above. My confidence is low in this score so I welcome thoughts from the authors and other reviewers.

**Questions:**

All questions stated above.

---

> ### Author Response · Authors · 2023-11-18
> **reply to reviewer feJt**
>
> Thank you for your significant engagement with our paper.   Despite a stated lack of expertise, we appreciated your perspective.
>
> > they say (method) is effective and efficient in a wide variety of domains including transport with constraints and unbalanced transport... For example, the constraints.. are on the marginal values themselves. However, what if the constraints come in a different form, e.g. a symmetry in the problem or distributions?
>
> We have been careful to only refer to 'marginal’ constraints ('intermediate’ or 'endpoint’) throughout the paper.   Symmetry constraints are fascinating, but beyond the scope of this work.
>
>
> > we can only observe population statistics and we want to infer individual statistics from this… But looking at the experiments, none of the experiments test this specific point. All the experiments are predicting population statistics.
>
> We agree that it would be interesting to test against ground truth individual trajectories.  Since this is not possible on real data, we constructed the leave-one-out experiments to test the quality of our marginal interpolations.
>
> > Below eq 27, the authors say that they perform optimization with respect to $\eta$ using a reparameterization trick. What is this trick? I couldn’t find a reference to it.
>
> The reparameterization trick refers to optimizing an expectation $E_{\rho}[f]$ under $\rho_t(x_t)$ by sampling some relevant stochasticity (here, $(x_0, x_1)$), parameterizing a transformation $x_t = g_\eta(x_0,x_1,t)$, and taking gradients using the chain rule, we have
>
> $$\frac{d}{d\eta}E_{\rho_t (x_{t})}\left[f\left(x_{t}\right)\right]=E_{\rho_0\left(x_{0}\right)\rho_1\left(x_{1}\right)}\left[\frac{\partial x_{t}}{\partial \eta}\frac{\partial f}{\partial x_{t}}\right]$$
>
> For more information, see: https://en.wikipedia.org/wiki/Variational_autoencoder#Reparameterization
>
> > Are the left-out marginals in the future and need to be extrapolated? Or is it interpolated as in-between points?
>
> We leave out one of the intermediate marginals each time we run our algorithm, and report the average W1 distance over runs.
>
> >The Madelung transformation in the appendix appears to only be for one particle… which is not of practical relevance.   The authors do not state this exactly
>
> The Madelung transformation in the appendix can be extended to $n$ particles by concatenating their coordinates into the vector $x \in \mathbb{R}^{dn}$, where $d$ is the dimensionality of the space, and introducing the corresponding symmetries (bosonic, fermionic) into the density and the vector field. The notorious challenges with the introduction of the symmetries, together with the fact that the Lagrangian becomes non-linear, suggest leaving the study of quantum systems beyond the scope of our paper.
>
> > The particular task in the experiment is tough to follow… e.g. simple things like what the raw data is in the form of. The resulting task is the PCA of this raw data.
>
> Thank you for the suggestion.   However, due to space constraints, we deferred details to Appendix and referenced related literature.
>
>
> >  I do not think this paper unifies the field. In fact, there were many situations where the authors stated to their credit that their method may not apply very well (e.g. Schrodinger equation). I am sure people would have to get a different perspective for these problems. So I would caution the authors in their language surrounding this point.
>
> We have tried to limit our use of 'unified’ to refer to our computational approach for the 'linearizable’ objectives in Sec. 3.2 and Eq. 17.   This level of generality is still notable, as many previous works have derived problem-specific methods which rely on additional assumptions or known structure in each setting.
>
> We feel that the perspective of ‘Lagrangian optimization on the density manifold’ and our novel dual objective in Thm 1 can be insightful to understand (and make connections between) a diverse range of problem settings, even where our computational approach does not directly apply.   For example, solving problems with nonlinear Hamiltonians, our dual objective suggests maintaining both a sampler and a density model to parameterize $\rho_t$.

---

> > ### Comment · Reviewer_feJt · 2023-11-22
> >
> > Thank you for the responses to my questions. I have a greater appreciation now for some of the challenges addressed in this work though my overall (admittedly limited) perspective remains the same. Given my lack of expertise, I will leave the ultimate decision up to the area chair and other reviewers.

---

### Official Review · Reviewer_zR9h · 2023-10-24

**Soundness:** 4 excellent
**Presentation:** 4 excellent
**Contribution:** 4 excellent
**Rating:** 8
**Confidence:** 3

**Summary:**

This paper introduces a deep learning based method to solve a wide range of Optimal Transport (OT) problems by using as the loss the dual of the Lagrangian obtained from the dynamic formulation of OT. It notably covers the Wasserstein distance, the Wasserstein-Fisher-Rao distance, physic constrained OT and Schrödinger bridges. The method is then applied on tasks of trajectory inference of single-cell RNA sequencing data.

**Strengths:**

The task of solving the dynamic OT problem is an important problem which has received recently much attention combined with deep learning. This works is part of this literature and provides a unified method which can be applied to several useful OT based problems.

- The paper provides a method applicable to solve different OT Problem by using the dual of the Lagrangian, which is a very nice contribution in my opinon.
- The theory and the method are well explained and a lot of examples are provided.
- While some part of the paper are rather technical, it is well written and comprehensive.
- The results on single-cell trajectory inference seem to be SOTA by combining different OT problems (namely UOT with physic constraints)

**Weaknesses:**

In my opinion, there is no major weakness. Maybe, a simple example showing how well the proposed method allows to approximate the true dynamic is lacking, for instance using the closed-form for the Schrödinger bridge between Gaussian proposed in [1].

[1] Bunne, Charlotte, Ya-Ping Hsieh, Marco Cuturi, and Andreas Krause. "The Schrödinger Bridge between Gaussian Measures has a Closed Form." In International Conference on Artificial Intelligence and Statistics, pp. 5802-5833. PMLR, 2023.

**Questions:**

I may have missed it, but I think the notation $\phi_{\overset{.}{\mu_t}}$, for instance in Equation (7), is not defined.

Typos:
- At equation (5), there is a point.
- Equation 23, a parenthesis is in the wrong place

---

> ### Author Response · Authors · 2023-11-18
> **reply to reviewer zR9h**
>
> Thank you for the positive review!   We are glad that you felt we explained technical content in a comprehensive and digestible way.
>
> We have included further experiments comparing with related SB methods which, in light of other reviewers' comments, we felt was higher priority than synthetic experiments.
>
> $\phi_{\dot{\mu}_t}$ indicates the cotangent vector corresponding to $\dot{\mu}_t$ as in Eq. 6 (which is written in terms of $\rho_t$ instead of $\mu_t$).

---

### Official Review · Reviewer_E9Se · 2023-11-01

**Soundness:** 3 good
**Presentation:** 2 fair
**Contribution:** 2 fair
**Rating:** 3
**Confidence:** 3

**Summary:**

The paper proposes a novel computational algorithm to obtain trajectories on the space of probability distributions that come from a Lagrangian variational formulation. The algorithm is based on a dual formulation of the problem, where the trajectories and dual functionals are represented with neural network. The algorithm is illustrated on several numerical examples, including the dynamics of a cell population, which also forms the main motivation for this paper.

**Strengths:**

The paper is concerned with an interesting problem that is valuable for research and applications.
The mathematical presentation is rigorous and educative.
The review of the literature is appropriate (to the best of my knowledge).

**Weaknesses:**

The paper does not provide sufficient evidence for the validity of their approach to solve all Lagranian type problems. The reported numerical experiments is limited and it is not clear what is the distingusihing characteristic of the proposed approach in relation to other works.

Minimizing the Lagrangian may not be the appropriate objective in general, because Lagrangian flows are not the minimizer, but the extreme points of the action integral.

Most of the background material can be moved to appendix as they are not critical to the proposed algorithm. The dual formulation, which is basically the main component, may be explained without much of the background material.

There is not enough motivation for parametrizing the flow according to (26). For a standard OT problem, what is the optimal function that replaces the neural net in this equation?

**Questions:**

Please see my comments above

---

> ### Author Response · Authors · 2023-11-18
> **reply to reviewer E9Se**
>
> Thank you for your review.   Below, we emphasize the `distinguishing characteristics’ of our approach and address other concerns.
>
> > it is not clear what is the distinguishing characteristic of the proposed approach in relation to other works.
>
> Our approach provides a novel method for solving optimal transport and its variants.
>
> Using a common parameterization and the objective in Eq. 17, our computational method applies to a diverse collection of ‘linearizable’ Lagrangians (Def 3.1).  Existing works often solve a single one of these problems (see Sec. 3.3) by leveraging known structure or particular assumptions (e.g. W2 OT solvers using ICNNs, Brenier Thm).
>
> Our main technical contribution is Proposition 1, which uses duality of tangent and cotangent space on the Wasserstein manifold to suggest computational objectives for solving Wasserstein Lagrangian Flows.
>
> > The paper does not provide sufficient evidence for the validity of their approach to solve all Lagranian type problems
>
> We agree that our computational approach does not apply for *all* Lagrangians, but applies for   `linearizable’ Lagrangians (defined in Sec. 3.2).   This includes W2 OT, physically constrained OT, OT with general Lagrangian costs, unbalanced OT, and entropic OT.
>
> We provide evidence for the effectiveness of these methods for solving trajectory inference problems in Sec. 4 and the experiments in the general response.
>
>
>
> > Minimizing the Lagrangian may not be the appropriate objective in general, because Lagrangian flows are not the minimizer, but the extreme points of the action integral.
>
> Optimal transport is explicitly defined via cost- or kinetic-energy minimization, and the existence of minimizers is guaranteed, for example, in the Benamou-Brenier formulation of W2 OT.   While the action functional could have multiple critical points (some of which are not minima), we know that all minimizers are critical points (i.e. Lagrangian flows).
>
>
> > There is not enough motivation for parametrizing the flow according to (26). For a standard OT problem, what is the optimal function that replaces the neural net in this equation?
>
> Thank you for the suggestion!   In the general response, we have argued that our parameterization in Eq. 26 can universally approximate any absolutely-continuous distributional path.    We have also defined the optimal function which the neural network approximates.
>
>
>
> > Most of the background material can be moved to appendix as they are not critical to the proposed algorithm
>
> Thank you for the feedback.
> In Sec. 2.2 and 2.3, we hope to introduce important concepts (kinetic energy, (co)tangent space) using familiar examples, before the more abstract and general presentation in Sec. 3.1-3.2.
> We will consider moving Sec. 2.1 to the appendix, where we give detailed consideration to OT with state-space Lagrangian costs (App. A.2.2).

---

> ### Comment · Reviewer_E9Se · 2023-11-22
>
> I appreciate the author's response to my questions.
>
> > Optimal transport is explicitly defined via cost- or kinetic-energy minimization, and the existence of minimizers is guaranteed, for example, in the Benamou-Brenier formulation of W2 OT. While the action functional could have multiple critical points (some of which are not minima), we know that all minimizers are critical points (i.e. Lagrangian flows).
>
> You are right about optimal transport, because it only contains the Kinetic energy.  But in the paper, you present the Lagranigian as difference of "Kinetic" - "potential" energies, which I assume it is inspired by classical Lagrangian mechanics. My point was to say that in Lagrangian mechanics, we are interested in the stationary points of the action integral, not minimizers.
>
> https://en.wikipedia.org/wiki/Stationary-action_principle
>
> > we have argued that our parameterization in Eq. 26 can universally approximate any absolutely-continuous distributional path.
>
> Thanks for your answer. I can now see how one can parameterize the optimal path in the way you did. But this representation does not seem very efficient as it requires a single neural net to represent a discontinuous function of x, and learning each of the potential functions \phi_0 or \phi_1 is difficult on their own.

---

> > ### Author Response · Authors · 2023-11-22
> >
> > > You are right about optimal transport, because it only contains the Kinetic energy. But in the paper, you present the Lagranigian as difference of "Kinetic" - "potential" energies, which I assume it is inspired by classical Lagrangian mechanics. My point was to say that in Lagrangian mechanics, we are interested in the stationary points of the action integral, not minimizers.
> >
> > Minimizers of the Lagrangian are the important critical/stationary points which correspond to the optimal solutions, even for *variants* of optimal transport (e.g. unbalanced OT [1, Eq 4-5],  physical or ground-space Lagrangian OT [2, Thm 7.21] , and Schrodinger Bridge [3, Problem 4.6]).
> >
> > In the current work, we are interested in incorporating prior knowledge about the state-space dynamics into the OT-like problems and thus focus on finding the minimizers of the Lagrangian.
> >
> > More generally, one could potentially find other critical points of the Lagrangian, but finding all possible critical points (stationary-action points) is not relevant to the setting of our paper.
> >
> > > this representation does not seem very efficient as it requires a single neural net to represent a discontinuous function of x
> >
> > We would like to highlight that we have addressed your initial concern regarding expressivity.
> >
> > We have demonstrated the effectiveness of the current parameterization in our experiments.   We make several additional points regarding the parameterization of $\rho_t$ and sampling procedure:
> >
> > - Note that in Algorithm 1, the sampling stage consists of two parts:   the neural network parameterized by $\eta$ (which, in theory, is expressive enough to match the correct marginals), and the non-parametric Wasserstein gradient updates according to Eq. 18. The non-parametric update significantly improved the efficiency of our sampling approach in practice.
> >
> > - Further, the neural network in Eq. 16 learns a continuous function in $t$, conditioned on a discontinuous input $t \leq ½$ or $t>½$ (and thus never learns a discontinuous function).  The design choice has found many applications in diffusion models [4].
> >
> > One could potentially design a more efficient sample-generating model, but we leave this for future work.
> >
> > ------
> > [1] Chizat et. al 2015.  “An Interpolating Distance between Optimal Transport and Fisher-Rao”
> >
> > [2] Villani 2008.   “Optimal Transport: Old and New” (Ch. 7)
> >
> > [3] Chen et. al 2020.  “Stochastic Control Liasons” (Problem 4.6)
> >
> > [4] Ho, J., & Salimans, T. (2022). “Classifier-free diffusion guidance”.

---

### Official Review · Reviewer_uk1R · 2023-11-02

**Soundness:** 3 good
**Presentation:** 3 good
**Contribution:** 3 good
**Rating:** 8
**Confidence:** 3

**Summary:**

This paper proposes a computational framework for solving Wasserstein
Lagrangian Flows. The focus is on problems in the form of equation (12)
where the cost is given by the minimum-action function in equation (1).
Special cases of this problem include OT, unbalanced OT,
and Schrodinger Bridge.
Proposition 1 shows the dual of this problem, and the main insight
of this paper is that it can be efficiently solved with a
flow given in equation (26).
This is done by the generative model in equation (26)
and solved with algorithm 1.
The experiments focus on single-cell RNA sequencing data.

**Strengths:**

1. One large challenge in developing OT and SB solvers in general
   and with flow-based models is needing to satisfy the
   marginal constraints.
   The generative model in equation (26) has the nice
   property that the marginals are always satisfied
   and the path is
2. Table 1 shows superior performances to OT, UOT, and SB methods.

**Weaknesses:**

1. My main concern with the paper (and perhaps the research area)
   is that the quality of the solution to the OT/UOT/SB problems
   is not directly measured or compared to other methods.
   In other words, the objective values of the optimization
   problem in (12) are never reported or compared between methods
   that are solving that problem.
   This is because solvers may solve the problem in different
   forms, such as dual forms, where the value being optimized
   is not comparable between methods.
   Because of this difficulty, research in this space often
   reports another diagnostic metric that is assumed to be
   correlated with a better solution to equation (12).
   In this paper, Table 1 reports the W1 distance to
   the left-out marginals. While they indeed show that
   WLF variants are often better than two other papers
   published in 2023, it would be better if the field
   had a more direct way of comparing solvers.
2. If I understand correctly, the representational capacity of equation (20) not
   clear or discussed. E.g., can it universally approximate distribution paths,
   and it is obvious that it can capture an optimal transport path?
   Experimentally it seems that it is the case, but it would be good to
   present or discuss more intuition behind this family of paths.
3. The experimental results are mostly focused on the
   existing scRNA setting. While these are seemingly good,
   it seems like the experimental settings considered
   by the other [Lagrangian SB](https://arxiv.org/abs/2204.04853)
   and [generalized SB](https://arxiv.org/abs/2209.09893)
   papers would be reasonable to compare against.
   This would help better-quantify the approach as
   a general-purpose solver for these large problem classes.
4. Appendix C.2 has some image generation experiments,
   but they seem incomplete and there are no quantitative
   evaluations of the method.

**Questions:**

I am overall positive about the methodological and
experimental contributions in this paper.
My only hesitation on the paper is in the weaknesses I pointed out
that 1) the paper never quantifies or compares the solution
quality of the OT/SB problems that are being solved
and 2) the expressivity of equation (20) is not clear to me (e.g., can
it universally approximate OT paths?).
I am very willing to re-evaluate my score during the
discussion period and would find it helpful if the
authors shared their thoughts.

---

> ### Author Response · Authors · 2023-11-18
> **reply to reviewer uk1R**
>
> Thank you for your review.  In particular, we feel the suggestions regarding Lagrangian SB experiments and expressivity of parameterization will improve the paper.
>
> > 1. My main concern with the paper (and perhaps the research area) is that the quality of the solution to the OT/UOT/SB problems is not directly measured
>
> Note that in Table 1-2, we compare our WLF-OT to the exact OT solution.  However, we agree that evaluation metrics are difficult in this research area.   Among scRNA setups, our experimental setting was motivated by (i) testing our ability to achieve accurate *interpolation* between given marginals and (ii) demonstrating how one might explore various Lagrangians or inductive biases for approximating ground-truth dynamics for a given problem of interest.
>
>
>
> > 2.  representational capacity of equation (20) not clear or discussed. E.g., can it universally approximate distribution paths, and it is obvious that it can capture an optimal transport path?
>
> Thank you for the suggestion!  In the general response, we have argued that our parameterization in Eq. 26 can universally approximate any absolutely-continuous distributional path (e.g. OT).   We will include a rigorous proof in the camera-ready.
>
> > 3. seems like the experimental settings considered by the other Lagrangian SB and generalized SB papers would be reasonable to compare against
>
> In the general response, we have included results in the experimental setting of the Lagrangian SB paper (which is also scRNA, but is their only non-toy experiment).  All of our proposed methods perform comparably or better than the best baseline method.
>
>
>
> > 4. Appendix C.2 has some image generation experiments, but they seem incomplete
>
> This is not a main focus of the work.   However, we note that exact OT requires the velocity vector field to be a gradient field, which is a more strict condition than alternative 1-step generation methods such as rectified flows.   In fact, exact OT methods have rarely been applied to high-dimensional settings such as images.    The generated images show that multi-step generation and one-step generations are nearly identical, which indicates satisfying the zero-acceleration condition.

---

> > ### Comment · Reviewer_uk1R · 2023-11-21
> >
> > Thanks for the response! I replied to the threads in the shared portions and think they are helpful clarifications. For this reason, I've updated my score from a marginal reject to an accept. To the best of my knowledge, the paper provides a reasonable methodological and experimental advancements for OT and single-cell trajectory modeling. I'm still hesitant that the experimental results do not directly measure the solution quality to the OT problems and are only on single-cell trajectory inference, but the experimental comparisons in the submission and response here seem reasonable and well-executed.

---

### Official Review · Reviewer_sAE6 · 2023-11-08

**Soundness:** 2 fair
**Presentation:** 2 fair
**Contribution:** 2 fair
**Rating:** 5
**Confidence:** 3

**Summary:**

This paper presents a novel deep learning-based framework for solving various variations of the optimal transport problem. Optimal transport involves finding the most efficient way to transport one distribution into another, taking into account different geometric and potential energy considerations. Realizations of this framework include the Schrödinger bridge, unbalanced optimal transport, and optimal transport with physical constraints, among others.
Solving the associated variational problems can be computationally challenging because the optimal density path is often unknown. However, the authors propose a dual formulation of the Lagrangians that leverages deep learning techniques to address these challenges. Importantly, their method does not require simulating or backpropagating through trajectories or accessing optimal couplings.

**Strengths:**

The paper introduces a novel deep learning-based framework for solving various variations of the optimal transport problem.
The proposed framework is versatile and can be applied to different variations of the optimal transport problem, such as the Schrödinger bridge, unbalanced optimal transport, and optimal transport with physical constraints. This approach offers a fresh perspective and provides a new ansatz to a challenging problem.

**Weaknesses:**

From a methodological standpoint, could the authors highlight the conceptual differences between your method and Chow et al. (2020) as well as Carrillo et al. (2021)? They seem to capture several of the frameworks discussed in this work and it is unclear in what sense you are providing a "unifying framework" here.

Besides, the current experimental validation is incomplete. Some questions:

1. I am surprised by the choices of baselines. Could the authors comment on why they only compared against Tong et al., 2023a and Tong et al., 2023b? There are several approaches that propose Schrödinger bridges for modeling single-cell dynamics and beyond, in particular Shi et al. (2023), Somnath et al. (2023), Liu et al. (2023), and Chen et al. (2022) to name a few. Vargas et al. (2021) and Somnath et al. (2023) in particular contain single-cell experiments.

2. It seems a bit odd not to compare against methods that also incorporate unbalanced solutions. Missing comparisons are here Eyring et al. (2022) and Lübeck et al. (2022) that propose neural unbalanced OT solvers. The most obvious comparison would be with Pariset et al. (2023) and Pooladian et al. (2023b), which you cite but do not compare against. Is there a reason why?

3. What is the motivation of running the method on the first 5 principal components (Table 1) besides running on 50 and 100 PCs? Does the method not scale to larger problems? Usually, single-cell data is reduced to 50 to 100 PCs to capture most of the variance, 5 PCs are not enough.

4. Why does the method barely outperform the exact OT method? For Table 2, why is exact OT even better on 50 and 100 PCS than the proposed method WLF-OT?

5. Is there any reasoning when to use which version of the method? If you compare the three methods using the same number of neural network parameters, do the results change?

6. In your conclusion you state: "we studied the problem of trajectory inference in biological systems, and showed that we can integrate various priors into trajectory analysis while respecting the marginal constraints on the observed data, resulting in significant improvement in several benchmarks". This is not what the results suggest in my opinion. The chosen datasets do not allow for a thorough study of cell death and birth and thus do not provide a framework that allows studying unbalanced formulations of OT. However, there are multiple datasets that contain screens of cancer drugs that induce cell death (see Srivatsan et al., 2020) as well as the datasets studied in Pariset et al. (2023), Lübeck et al. (2022), or Eyring et al. (2022) that would be more suitable.

7. Minor, but a work connected to "Optimal Transport with Lagrangian Cost" to include is Fan et al. (2022).

References:

- Shui-Nee Chow, Wuchen Li, and Haomin Zhou. Wasserstein Hamiltonian flows. Journal of Differential Equations, 268(3):1205–1219, (2020).
- Carrillo, Jose A., Daniel Matthes, and Marie-Therese Wolfram. "Lagrangian schemes for Wasserstein gradient flows." Handbook of Numerical Analysis 22 (2021).
- Shi, Y., De Bortoli, V., Campbell, A. & Doucet, A. Diffusion Schrödinger Bridge Matching. arXiv preprint arXiv:2303. 16852 (2023).
- Liu, Guan-Horng, et al. "I$^ 2$SB: Image-to-Image Schrödinger Bridge." International Conference on Machine Learning (ICML) (2023).
- Somnath, Vignesh Ram, et al. "Aligned Diffusion Schrödinger Bridges." Conference on Uncertainty in Artificial Intelligence (UAI) (2023).
- Vargas, Francisco, et al. "Solving Schrödinger bridges via maximum likelihood." Entropy 23.9 (2021).
- Chen, Tianrong, Guan-Horng Liu, and Evangelos A. Theodorou. "Likelihood training of Schrödinger bridge using forward-backward SDEs theory." International Conference on Learning Representations (ICLR) (2022).
- Lübeck, F. et al. Neural unbalanced optimal transport via cycle-consistent semi-couplings. arXiv preprint arXiv:2209.15621 (2022).
- Eyring, Luca Vincent, et al. "Modeling single-cell dynamics using unbalanced parameterized Monge maps." bioRxiv (2022).
- Pariset, Matteo, et al. "Unbalanced Diffusion Schrödinger Bridge." arXiv preprint arXiv:2306.09099 (2023).
- Srivatsan, S. R. et al. Massively multiplex chemical transcriptomics at single-cell resolution. Science 367, 45–51 (2020).
- Fan, Jiaojiao, et al. "Scalable computation of Monge maps with general costs." ICLR Workshop on Deep Generative Models for Highly Structured Data (2022).

**Questions:**

No further questions.

---

> ### Author Response · Authors · 2023-11-18
> **reply to reviewer sAE6**
>
> We appreciate the knowledgeable and thorough review!   We have run two sets of new experiments to address the concern about additional baselines, and summarized our results in the general response (and soon in the new revision). We address the remaining concerns below.
>
> > highlight the conceptual differences between your method and Chow et al. (2020) as well as Carrillo et al. (2021)?
>
> Comparison with Chow et al. (2020) was discussed in detail at the top of page 9 in the original submission.  Carrillo et al. (2021) is concerned with JKO-style discretizations of W2 gradient flows, which does not address endpoint-constrained problems such as optimal transport.
>
>
> > 1. several approaches that propose Schrödinger bridges for modeling single-cell dynamics and beyond
>
> In the general response, we have added comparison with additional SB baselines (Vargas et. al 2021, Chen et. al 2022 (FB-SDE), de Bortoli et. al 2022, along with Koshizuka et. al 2023 (Neural Lagrangian SB) and others).
>
>
> > 2. It seems a bit odd not to compare against methods that also incorporate unbalanced solutions… the most obvious comparison would be with Pariset et al. (2023) and Pooladian et al. (2023b), which you cite but do not compare against. Is there a reason why?
>
> Thank you for the suggestion. We have included the comparison with Pariset et al. (2023) in our general response.
>
> Note, Pooladian et. al 2023b is not an unbalanced transport method.  Their Sec. 5.1 considers physically-constrained OT, but only in two-dimensional experiments.
>
> > 3. First 5 principal components (Table 1).  Does the method not scale to larger problems?
>
> We demonstrate the scalability of our method in Table 2 in the original submission, showing results on 50 and 100 principal components for the Cite and Multi datasets.   As in the 5-d cases, our proposed approach achieves improvements over the baselines.
>
>
>
> > 4. Why does the method barely outperform the exact OT method? For Table 2, why is exact OT even better on 50 and 100 PCS than the proposed method WLF-OT?
>
>
>   WLF-OT and OT-CFM approximate the straight-path exact OT solution (which is not differentiable when transitioning between transportation plans) with a differentiable function.  This may lead to better performance in some cases, since the exact OT solution does not reflect the ground-truth dynamics of the cells in either Table 1 or 2.
>
> We also note that the evaluation metric in this setting is the W1 distance rather than W2, in keeping with previous works on the subject (Tong et. al 2023a,b, Koshizuka et. al 2023).
>
>
>
>
>
> > 5.  Is there any reasoning when to use which version of the method?
>
> Our goal was to emphasize throughout the paper (Introduction, first sentence of each Example in Sec. 3.3, Conclusion) that prior knowledge or inductive bias of the ground truth dynamics of the task should inform the choice of Lagrangian.
>
> > 5.  If you compare the three methods using the same number of neural network parameters, do the results change?
>
>
> The results for our proposed WLF methods use MLPs with the same number of layers and neurons.
> For baseline methods, the differences in architectures or number of parameters are insignificant from the perspective of the runtime.
>
>
> > 6. "we studied the problem of trajectory inference in biological systems, and showed that we can integrate various priors into trajectory analysis while respecting the marginal constraints on the observed data, resulting in significant improvement in several benchmarks". This is not what the results suggest in my opinion.  The chosen datasets do not allow for a thorough study of cell death and birth / unbalanced OT.
>
> We believe this statement is supported by our exposition and results, and does not make specific claims about unbalanced OT or birth/death dynamics.   As requested, we have included an additional unbalanced OT experiment capturing birth/death dynamics in the general response.

---

### Author Response · Authors · 2023-11-18
**general response**

To spur discussion, we provide an initial response to several points which were common across reviewers, and will post a new revision soon.

* We provide experiments showing improvement of our proposed methods over numerous additional SB (sAE6, uk1R), and unbalanced OT baselines  (sAE6).

* In the second general response, we argue for the expressivity of NN parameterization of the distributional path $\rho_t$ (uk1R, zR9h, E9Se).  We refer to reviewers with their (score, confidence).

**Experiment:  Lagrangian SB Setting with SB Baselines**

Reviewer sAE6 (5,3) requested additional baselines among SB methods and unbalanced OT, while reviewer uk1R (5,3) suggested considering additional experimental settings, such as those in (Koshizuka et. al 2023)[2]. We test our method on the task in Table 1 of [2] below,  and find that all of our proposed methods perform similarly or better than baseline methods.  We show improvement over related SB work in IPF(GP) [4] , IPF(NN) [5], SB-FBSDE [6], among others.

| Model (W1 ↓)        | t1   | t2   | t3   | t4   |
|-------------------------|------|------|------|------|
| Neural SDE              | 0.69 | 0.91 | 0.85 | 0.81 |
| OT-Flow + OT            | 0.85 | 1.05 | 1.09 | 1.00 |
| TrajectoryNet           | 0.73 | 1.06 | 0.90 | 1.01 |
| IPF (GP) [4]               | 0.70 | 1.04 | 0.94 | 0.98 |
| IPF (NN) [5]               | 0.73 | 0.89 | 0.84 | 0.83 |
| SB-FBSDE [6]               | 0.56 | 0.80 | 1.00 | 1.00 |
| NLSB (E+D+V) [2]           | 0.68 | 0.84 | 0.81 | 0.79 |
| WLF-OT                  | 0.65 | 0.78 | 0.76 | 0.75 |
| WLF-SB                  | 0.63 | 0.79 | 0.77 | 0.74 |
| WLF-(OT + potential)    | 0.64 | 0.77 | 0.76 | 0.76 |
| WLF-UBOT                | 0.64 | 0.84 | 0.80 | 0.81 |
| WLF-(UBOT + potential)  | 0.67 | 0.80 | 0.78 | 0.78 |

*Task Description*:   We reproduce the setting from [2] exactly by taking the dataset from their repository with the same train/test split.   The data is the same as the EB column in our Table 1 (5-dimensional PCA as data). However, the model is trained on samples from all time marginals using an 80/20 train/test split. For evaluation, held-out test samples from $\mu_{t_{i-1}}$ are propagated to $\rho_{t_i}$ and compared to the test samples from $\mu_{t_i}$ using the W1 distance.

**Experiment: Comparison with Unbalanced OT Baseline**:

As requested by Reviewer sAE6 (5,3), we compare our method to Pariset et. al 2023, which considers an unbalanced setting. Both our optimal transport and unbalanced transport algorithms outperform the competitors.

| Method                      | MMD   (↓)    | W2 (↓)   |
|-----------------------------|-----------|------|
| Chen et al. (2021) [6]        | 1.86e-2   | 6.23 |
| Pariset et al., no deaths/births | 1.86e-2   | 6.27 |
| Pariset et al. [3]             | 1.75e-2   | 6.11 |
| WLF-OT                      | 5.04e-3   | 5.20 |
| WLF-UBOT                    | 9.16e-3   | 5.01 |

*Task Description*: We use the dataset from Pariset et al., which tracks the time evolution of single-cell markers of melanoma cells undergoing treatment with a cancer drug treatment for two different drugs. The dataset consists of three marginals, which they split 80/20 train/test split for every marginal. The model is trained on the train marginals and then is evaluated by propagating the samples from the first test marginal and comparing them to the last test marginal using Wasserstein-2 and MMD metrics. We use the train/test data from their repository to make sure that the setting is the same. For unbalanced algorithms, to evaluate MMD we sample proportionally to the weights without replacement.

[1] Ambrosio et. al 2006, “Gradient Flows in Metric Spaces and in the Space of Probability Measures”

[2]  Koshizuka et. al 2023 “Neural Lagrangian Schrodinger Bridge”

[3] Pariset et al 2023, “Unbalanced Diffusion Schrodinger Bridge”

[4]  Vargas et. al 2021 “Solving Schrödinger Bridges via Maximum Likelihood”

[5]  de Bortoli et. al 2021 “Diffusion Schrödinger Bridge”

[6]  Chen et al 2021, “Likelihood Training of Schrödinger Bridge using Forward-Backward SDEs Theory”

---

> ### Author Response · Authors · 2023-11-18
> **expressivity of parameterization**
>
> **Parameterization of Distributional Path $\rho_t$**
>
> Reviewers uk1R (5,3) and zR9h (8,3) asked about the expressivity of our parameterization, while E9Se (3,3) asked about its motivation and optimality conditions.   We argue below that our parameterization can universally express absolutely-continuous distributional paths $\rho_t$, and agree with reviewers that this result will improve our revised submission.
>
> First, consider the W2 optimal transport case, where (given the optimal $\phi^*$) one can sample $x_t \sim \rho_t^*$ from the optimal marginals using $x_0 \sim \mu_0$ and $x_t =x_0 + t \nabla \phi_0^*(x_0)$, or $x_1 \sim \mu_1$ and $x_t =x_1 - (1-t) \nabla \phi_1^*(x_1)$ by the Brenier Thm. Consider a function $f: (x_0, x_1, t, \mathbb{I}[t < .5]) \mapsto \begin{cases} x_0 + t \nabla \phi_0^*(x_0) \qquad \text{ if } t<0.5    \newline x_1 - (1-t) \nabla \phi_1^*(x_1) \text{ if } t \geq 0.5
> \end{cases} $.
>
> The NN in our parameterization in Eq. 26 has the same signature as $f$ above.  Since we also pass an indicator variable  $\mathbb{I}[t < .5]$ to our neural network (as described below Eq. 28), the NN is able to learn a discontinuous function.   By the universal expressivity of neural networks, our parameterization is thus expressive enough to learn the optimal $f$ ( after accounting for the $(1-t)x_0 + t x_1$ term), which yields samples $x_t \sim \rho_t^*$ from the optimal marginals for the W2 OT problem.
>
> More generally, any absolutely-continuous distributional path $\rho_t$ on the W2 manifold can be traced using an ODE with a vector field $v_t^* = \nabla \phi^*_t$ (Ambrosio et. al 2006 Thm 8.3.1 [1]).   Following similar reasoning as above, we define the function $f$ by running the ODE in the appropriate direction, e.g. $x_t = ODE(x_0; \phi^*_t)$ if $\mathbb{I}[t < .5]$.

---

> > ### Comment · Reviewer_uk1R · 2023-11-21
> >
> > Quickly commenting in here to say this is really nice! I am now convinced the model is capable of capturing the true OT path while having the great property of preserving the marginals. Before, I did not realize that it was so easy to parameterize a probability path/stochastic interpolant with a neural network that can capture the OT path.

---

> > > ### Author Response · Authors · 2023-11-22
> > >
> > > Thank you for your constructive feedback, which improved our paper!

---

> ### Comment · Reviewer_uk1R · 2023-11-20
> **Clarification on evaluation metric on the Lagrangian SB settings**
>
> > For evaluation, held-out test samples are propagated and compared to the test samples using the **W1 distance**.
>
> Table 1 of the Lagrangian SB paper [2] uses the W2 distance, not the W1 distance. Is it possible that the baseline methods in the table you sent use the W2 distance as they were reported in [2], but the WLF results are measured using the W1 distance?
>
> (Also, it would be useful to include the NLSB results in the final version of the table as well)

---

> > ### Author Response · Authors · 2023-11-20
> >
> > Thank you for your comment!
> > - For the evaluation, we followed closely their [implementation](https://github.com/take-koshizuka/NLSB/blob/b1ac92b81c844184f72d5cb973611888928c057a/utils.py#L112), hence the comparison is correct.
> > - The best-performing NLSB method is present in the table on the 7th row. We will include the tables in the final version of the paper.
> >
> > Let us clarify the potential confusion of the notation. According to [this definition](https://en.wikipedia.org/wiki/Wasserstein_metric#Definition), Lagrangian SB uses $W_1$ with Euclidean distance $d(x,y)= || x-y ||_2$
> >
> > $$W_p = \left(\inf_\pi \int d(x,y)^p ~ \pi(x,y) dxdy \right)^{1/p}.$$
> >
> > In the paper, the authors refer to the metric as EMD-L2, which matches $W_{p=1}$ with the Euclidean distance (our notation). Indeed,
> > $$\text{EMD-L2} = \inf_\pi \int ||x - y||_2 ~ \pi(x,y) dxdy.$$

---

> > > ### Comment · Reviewer_uk1R · 2023-11-21
> > >
> > > Thank you for the clarification! This part seems correct then. I was a little confused earlier because page 7 of the LSB paper says that  `EMD-L2` is "*the square root of the earth mover’s distance with L2 cost*", but it appears the code does not include the square root and it is the standard W1 distance, as you define.

---

### Author Response · Authors · 2023-11-22
**revision**

Thank you again to the reviewers for valuable comments.    We have posted an updated revision which reflects changes suggested during the review and rebuttal period.  In particular,

- We have included the experiments from the general response in Sec. 5, which show that our method can outperform a number of Schrodinger Bridge and unbalanced OT baselines (as requested by sAE6 and uk1R)

- We have added a proposition in Sec. 3.2.2 (Proposition  2 in the new revision), which establishes the ability of our parameterization to represent any (continuous) distributional path $\rho_t$ (as requested by uk1R, zR9h, E9Se)

- We also improved the exposition and discussions of related works.

---

### Meta-Review · Area_Chair_H2JG · 2023-12-08

**Metareview:**

The authors propose a novel deep learning algorithmic approach to solve various optimal transport (OT) problem including Schrodinger bridge, unbalanced OT, OT with physical constraints. To overcome the challenge on the unknown optimal density path, the authors propose a dual formulation of the Lagrangians and solve it using deep learning approach. The proposed method does not need the simulating or backpropagating through trajectories, or accessing optimal couplings. The proposed ideas are interesting. However, the Reviewers raised several concerns about experiments, e.g., datasets, baselines, setups, evaluation metrics. The paper has a mixed comments from the Reviewers. Although the authors provide additional experimental results during the rebuttal, there are still concerns from the Reviewers about solution quality to the OT problem, more comprehensive experiments w.r.t. baselines, datasets, evaluation metrics. In brief, I think another round of review is necessary. The authors may follow the Reviewers' comments to improve the work.

**Justification For Why Not Higher Score:**

+ There are still concerns from the Reviewers about solution quality to the OT problem, more comprehensive experiments w.r.t. baselines, datasets, evaluation metrics (even with the additional experimental results in the rebuttal). So, it seems the experiments may not be comprehensive to support the proposed approach yet. Thus, I think another round of review is necessary.

**Justification For Why Not Lower Score:**

N/A

---

### Decision · Program_Chairs · 2024-01-16

Reject